



# Snow Water Equivalents exclusively from Snow Heights and their temporal Changes: The ΔSNOW.MODEL

Michael Winkler[1,*], Harald Schellander[1,2,*], and Stefanie Gruber[1]

[1]ZAMG – Zentralanstalt für Meteorologie und Geodynamik, Innsbruck, Austria
[2]Department of Atmospheric and Cryospheric Sciences, University of Innsbruck, Austria
[*]These authors contributed equally to this work.

**Correspondence:** Michael Winkler (michael.winkler@zamg.ac.at) and Harald Schellander (harald.schellander@zamg.ac.at)

**Abstract.** Snow heights have been manually observed for many years, sometimes decades, at various places. These records are often of good quality. In addition, more and more data from automatic stations and remote sensing are available. On the other hand, records of snow water equivalent $SWE$ – synonymous for snow load or mass – are sparse, although it might be the most important snowpack feature in fields like hydrology, climatology, agriculture, natural hazards research, etc. $SWE$ very often has to be modeled, and those models either depend on meteorological forcing or are not intended to simulate individual $SWE$ values, like the substantial seasonal "peak $SWE$".

The ΔSNOW.MODEL is presented as a new method to simulate local-scale $SWE$. It solely needs snow heights as input, though a gapless record thereof. Temporal resolution of the data series is no restriction per se. The ΔSNOW.MODEL is a semi-empirical multi-layer model and freely available as R-package. Snow compaction is modeled following the rules of Newtonian viscosity. The model considers measurement errors, treats overburden loads due to fresh snow as additional unsteady compaction, and melted mass is stepwise distributed top-down in the snowpack.

Seven model parameters are subject to calibration, which was performed using 71 winters from 14 stations, well-distributed over different altitudes and climatic regions of the Alps. Another 73 rather independent winters act as validation data. Results are very promising: Median bias and root mean squared error for $SWE$ are only $-4.0\,\mathrm{kg\,m^{-2}}$ and $23.9\,\mathrm{kg\,m^{-2}}$, and $+2.3\,\mathrm{kg\,m^{-2}}$ and $23.1\,\mathrm{kg\,m^{-2}}$ for peak $SWE$, respectively. This is a major advance compared to snow models relying on empirical regressions, but also much more sophisticated thermodynamic snow models not necessarily perform better.

Not least, this study outlines the need for comprehensive comparison studies on $SWE$ measurement and modeling at the point and local scale.

## 1 Introduction

Total height ($H$) and bulk density ($\rho_\mathrm{b}$) are fundamental characteristics of a seasonal snowpack (e.g., Goodison et al., 1981; Fierz et al., 2009). Equation (1) links them to the areal density $[\mathrm{kg\,m^{-2}}]$ of the snowpack, which – in hydrological applications



– is usually referred to as snow water equivalent ($SWE$), as it resembles "the depth of water that would result if the mass of snow melted completely" (Fierz et al., 2009).

$$SWE = H \cdot \rho_{\mathrm{b}} \qquad [1\,\mathrm{kg\,m^{-2}} \equiv 1\,\mathrm{mm\,water\,equivalent\,(w.e.)}] \tag{1}$$

## 1.1 Measurements of $H$ and $SWE$

Measuring $H$ is relatively easy (e.g., Sturm and Holmgren, 1998): Manual measurements at a certain point only require a rod or ruler (e.g., Kinar and Pomeroy, 2015), and decades-long series of daily $H$ measurements exist in many regions – in lowlands as well as in alpine areas (e.g., Haberkorn, 2019). In modern times more and more $H$ data from automated measurements (mostly sonic or laser) become available, typically in sub-hourly resolution (McCreight and Small, 2014). In addition, remote sensing techniques currently increase the number of $H$ data significantly, having the advantage of an areal picture instead of point information but at the cost of accuracy and – in most cases – temporal resolution and regularity (Cf., e.g., Dietz et al. (2012) for a general review of methods, and Deems et al. (2013) for a review on lidar measurements. Painter et al. (2016) provide a thorough overview. Garvelmann et al. (2013) and Parajka et al. (2012), e.g., illustrate the potential of timelapse photography.)

In contrast, measurements of $SWE$ (or $\rho_{\mathrm{b}}$) are more difficult (e.g., Sturm et al., 2010): Manual measurements are time consuming (especially if a snow pit is dug), require some basic equipment like snow tubes or snow sampling cylinders and – not least – some dexterity (e.g., Kinar and Pomeroy, 2015). As a consequence, $SWE$ measurements are carried out at much fewer locations than $H$ measurements (e.g., Mizukami and Perica, 2008; Sturm et al., 2010), their accuracy is lower, and series are shorter. Only in very rare cases consecutive, decades-long measurement series are available (e.g., in Switzerland; cf. Jonas et al., 2009). Often they are only carried out at irregular intervals ("snow courses") and even if regularly measured, temporal resolution is hardly ever higher than two weeks. Besides these restrictions concerning manual measurements, also automatic measurements of $SWE$ are not at all comparable in quality and quantity with automated $H$ measurements. They are quite expensive, often inaccurate, still at a developmental stage, and/or suffer from significant problems if not intensively maintained throughout the snowy season. Methods involve weighing techniques (snow scales; e.g., Smith et al., 2017; Johnson et al., 2015), pressure measurements (snow pillows; e.g., Goodison et al., 1981), upward-looking ground penetrating radar (GPR; e.g., Heilig et al., 2009), passive gamma radiation (e.g., Smith et al., 2017), cosmic ray neutron sensing (CRNS; e.g., Schattan et al., 2019), L-band Global Navigation Satellite Signals (GNSS; e.g., Koch et al., 2019), etc. Presumably, the biggest and best serviced network of automated $SWE$ measurements is SNOTEL with about 800 sites in Western North America (Avanzi et al., 2015).

Furthermore, there is no way to directly monitor $SWE$ by remote sensing techniques (Schaffhauser et al., 2008), and deriving this snow property from satellite products at the local scale ($< 1\,\mathrm{km}$) is still not possible (Smyth et al., 2019). On top of that, there is the issue of longterm availability: automated measurements and (at least rough) remote sensing of $SWE$ have not been available for more than some twenty years at their best (e.g., SNOTEL, operated since the late 1990s), a fairly short timespan compared to decades-long daily $H$ data (e.g., Kinar and Pomeroy, 2015).





Regardless of these problematic circumstances accompanying $SWE$ measurements, many hydrological and agricultural
applications depend on good estimates of $SWE$ (e.g., Goodison et al., 1981; Sturm and Holmgren, 1998). Very often, in the
end the mass of water stored in the snowpacks matters, and that's why the majority of those fields is especially interested
in seasonal $SWE$ maxima, i.e. "peak $SWE$" ($SWE_{\mathrm{pk}}$). $SWE_{\mathrm{pk}}$ are also the main focus of different kinds of extreme value
and climatic analyses, both of which additionally very much rely on longterm or even "historical" data. Not least, snow load
standards (e.g., International Organization for Standardization, 2013) rely on extreme value analyses of longterm snow load
records (or $SWE$ records, respectively; snow load $SL = SWE \cdot g$, with gravitational acceleration $g = 9.8\,\mathrm{m\,s}^{-2}$). These points
reveal the great discrepancy between the good data situation in terms of $H$ on the one hand, and the insufficient availability of
$SWE$ data on the other.

## 1.2  Modeling $SWE$

Modern snow models like Crocus (e.g., Vionnet et al., 2012), SNOWPACK (e.g., Lehning et al., 2002), SNTHERM (Jor-
dan, 1991), or dual-layer model SNOBAL (Marks et al., 1998) resolve mass and energy exchanges within the ground-snow-
atmosphere regime in a detailed way by depicting the layered structure of seasonal snowpacks. (Echoing Langlois et al. (2009),
these models will be termed "thermodynamic snow models" in the following.) Estimations of absolute model errors for $SWE$
are rather scarce. However, it might be at the order of $10\,\mathrm{to}\,15\,\mathrm{kg\,m}^{-2}$ (Langlois et al., 2009), but probably sometimes sig-
nificantly more (Vionnet et al., 2012). See Sect. 3.2 and Table 2 for more details. All thermodynamic snow models need
meteorological input data like temperature or radiation. Unfortunately, many longterm $H$ series – which are so valuable for a
variety of applications (see above) – do not come along with these data, and parametrizing or downscaling them from other
sources in turn isn't straightforward either. Even markedly simpler models at least require temperature and precipitation mea-
surements as inputs (e.g., De Michele et al., 2013) or – in a very recent work by Hill et al. (2019) – climatological means
thereof. Consequently, no thermodynamic snow model is applicable to derive $SWE$ exclusively from $H$. (Avanzi et al. (2015)
provide a good review and also introduce models including "statistical descriptions" which, in turn, need $SWE$ measurements
as input and, therefore, are not further addressed here. In this study these are counted as "thermodynamic snow models" as
well.)

On the other side of the $SWE$ modeling spectrum there are those models which – aside $H$ – only depend on date $d$ (Pis-
tocchi, 2016), $d$ and altitude $z$ (Gruber, 2014, see statistical approach therein), $d$ and regional parameters (e.g., Mizukami and
Perica, 2008; Guyennon et al., 2019) or $d$, $z$ and regional parameters (e.g., Jonas et al. (2009); and applications thereof e.g.,
by Achleitner and Schöber (2017), who fitted the parameters to Austrian data). Again, Avanzi et al. (2015) provide a thorough
listing of those models, which will be abbreviated as ERMs (for "empirical regression models") in the following. ERMs very
much rely on the strong, near-linear dependence between $H$ and $SWE$ (cf., e.g., Jonas et al., 2009). According to Gruber
(2014) and Valt et al. (2018) $H$ describes more than 80% of $SWE$ variance (81% and 85%, respectively). This behavior bases
on the narrow range within which the majority of bulk snow densities is found, and it leads to the well-known characteristic
of $H$–$SWE$–$\rho_{\mathrm{b}}$ datasets: log-normally distributed $H$ and $SWE$ as well as normally distributed $\rho_{\mathrm{b}}$ (e.g., Sturm et al., 2010).
Unfortunately, ERMs cannot adequately model (unchanged) $SWE$ during periods with snow densification only due to meta-





morphism and deformation strain (Jordan et al., 2010) but without (significant) mass loss. Sturm and Holmgren (1998) already state that $H$ and "load" (or $SWE$, respectively) play a more limited role in determining the compaction behavior in seasonal

snow than "grain and bond characteristics" and temperature.

Interestingly, in ERMs absolute, single-day $H$ observations are the only snow characteristics used. Depending on calibration focus they either adequately model mean $SWE$ or $SWE_{\mathrm{pk}}$, mid winter or spring, etc. This is an inherent fact due to their model architecture. Those calibrated for good estimates of mean $SWE$ (necessarily) fail to model $SWE_{\mathrm{pk}}$ sufficiently well, those designed for $SWE_{\mathrm{pk}}$ often give bad $SWE$ results during phases with shallow snowpacks. Typically, they simulate unrealistic

mass losses during phases with compaction only by metamorphism and deformation strain, and the timing of $SWE_{\mathrm{pk}}$ as well as the duration of high snow loads cannot be modeled well. As it is honestly stated by Jonas et al. (2009) – the authors of one of the most influential ERMs – those models cannot be used to "convert time series of $H$ into $SWE$ at daily resolution or higher" because they may "feature an incorrect fine structure in the temporal course of $SWE$". Therefore, ERMs are not suitable to calculate $SWE$ for individual days. (Still, well made for means.)

McCreight and Small (2014) go an interesting step further and not only use single-day $H$ values for their regression model, but also the "evolution" of daily $H$. They make use of the negative correlation of $H$ and $\rho_{\mathrm{b}}$ at short timescales (10 days) and their positive/negative correlation at longer timescales (3 months) during accumulation/ablation phases. This promising step of development is limited by the fact that the model parameters can only be estimated through regressions relying on at least three training datasets of $H$ and $\rho_{\mathrm{b}}$ (or $SWE$) from nearby stations. (McCreight and Small (2014) used ultrasonic $H$ measurements

in conjunction with $SWE$ pillow measurements.) Unfortunately, this disqualifies the model of McCreight and Small (2014) for assigning $SWE$ to longterm and historical $H$ series as consecutive $SWE$ measurements are not available for those.

Searching for alternative approaches that link $H$ and $SWE$ throughout a snowy season without the need of meteorological input like temperature one might find Martinec and Rango (1991)'s work which provides a semi-empirical model that – in some respect – bridges the gap between thermodynamic models (needing lots of meteorological input) and ERMs (being

"overregulated" by snow height). They use a method already developed by Martinec (1956) "to compute the water equivalent from daily total depths of the seasonal snow cover". Snow compaction is expressed as a time-dependent power function, and they end up with the following equation for each layer's snow density $\rho_n$ after $n$ days

$$\rho_n = \rho_0 \cdot (n+1)^{0.3}, \tag{2}$$

where $\rho_0$ is the initial density of the snow layer. Martinec and Rango (1991) use a constant $\rho_0$ of $100\,\mathrm{kg\,m^{-3}}$ and a fixed

exponent of 0.3, without going into detail how these values were found. They show that the error made by a bad choice of $\rho_0$ is rapidly decreasing with $n$ and therefore the power function (Eq. (2)) gives robust results at least for old(er) snow layers. They claim (without further explanation) that snow "luckily" does not settle according to an exponential curve, and show that in that case the error of $\rho_0$ would be independent from $n$ and would not diminish while the snow layers are aging. Their model interprets "each increase of total snow depth [. . . ] as snow fall" and "[if] the total snow depth remains higher than the settling

by [Eq. (2) of the article in hand], this is also interpreted as new snow. If the snow depth drops lower than the value of the





superimposed settling curve of the respective snow layers, it is interpreted as snowmelt, and a corresponding water equivalent is subtracted. In this way the water equivalent of the snow cover can be continuously simulated [. . . ]". Martinec and Rango (1991) get promising results using this simple but robust snow compaction law. They only need daily snow heights as input and end up with a modeled $SWE$ record at daily resolution.

## 1.3 Motivation for a New Approach

The question evolves, whether such a semi-empirical, layer-resolving snow model like the rather old one of Martinec and Rango (1991) can be improved and modernized, in order to provide an up-to-date snow model standard somewhere between sophisticated, thermodynamic and modest, purely-statistical $H$–$SWE$–$\rho_b$ models, like ERMs. Looking at the ease of Martinec and Rango (1991)'s approach, thinking about modern computational possibilities, and given the introductorily described strong need for "something handy", it seems interesting that there are no recent publications on this topic. This leads to a possible null hypothesis of the paper-in-hand: "It is not possible to better model snow water equivalents than empirical regression models do, by exclusively using snow heights and their temporal changes as input." This statement can be rejected after the following presentation of a new way to assign $SWE$ values to the huge number of longterm, historical, and high-quality $H$ records of daily resolution. It follows Martinec and Rango (1991)'s key feature of considering daily *change* of snow height as a proxy for the various processes altering bulk snow density $\rho_b$ and snow water equivalent $SWE$, but further

- bases its (dry) snow densification function on Newtonian viscosity,

- provides a way to deal with small discrepancies between model and observation (in the order of $H$ measurement errors),

- takes into account unsteady compaction of underlying, older snow layers due to overburden snow loads,

- densifies snow layers from top to bottom during melting phases without automatically modeling mass loss due to runoff,

- and offers a way to deal with rain on snow (as an optional step, which is not detailed yet).

This new modeling approach is named ΔSNOW.MODEL and an easy-to-use R-package is available through https://r-forge. r-project.org. The package is called *nixmass*, and it not only involves the ΔSNOW.MODEL, but also other models that use snow height (*nix*. . . Latin for "snow") to simulate $SWE$ (i.e., snow *mass*).

The ΔSNOW.MODEL neither gives any new crucial insights in snow physics nor involves substantially new approaches. Still, the ΔSNOW.MODEL "rearranges" existing components in a physically consistent way and – as a whole – represents a new *method*. That is why it is described in the following Method section of this publication (Sect. 2). The calibration is outlined in Sect. 2 as well. Results, like best parameter choices and validation of the model output when compared to measurements, are given in Sect. 3. Section 4 provides an application of the ΔSNOW.MODEL for spatially modeling extreme snow loads in Austria. Sections 5 and 6 discuss possible future developments and provide concluding remarks.





## 2 Method

As a successor of Martinec and Rango (1991)'s model the $\Delta$SNOW.MODEL also builds on a semi-empirical approach and, therefore, can be regarded as standing between thermodynamic and empirical regression models. Its basic version was already presented by (Gruber, 2014, chapter 4) and a revision was presented by Gruber et al. (2018), but the $\Delta$SNOW.MODEL described in the present article experienced a significant updating and recoding. It is designed for seasonal snowpacks and not intended for (multiannual) firn. The $\Delta$SNOW.MODEL does not need any further input but a gapless snow height record. It uses changes in observed snow height to simulate a record of $SWE$. A scheme of the $\Delta$SNOW.MODEL's principle is shown in Fig. 1.

Snow compacts over time due to various processes. Jordan et al. (2010) categorize them in snow drift, metamorphism, and deformation strain. The $\Delta$SNOW.MODEL cannot deal with snow drift, however, it differentiates between the latter two processes. Metamorphism and deformation strain are processed in the three modules *Dry Metamorphism* (2.1), *Fresh Snow* (2.2.1), and *Drenching* (2.2.3). The fourth module, *Scaling* (2.2.2), accounts for small discrepancies between model and observations.

### 2.0.1 Preliminary: The First Snow Layer

For non-zero snow height observations ($H_{obs} > 0$) after a snow-free period the $\Delta$SNOW.MODEL assigns the following features to the model snowpack: There is one snow layer (layer counter $ly = 1$) and the age of this layer is set to $age = 1$. Snow height of this model layer ($h$) and total model snow height ($H$) are equal, and set to observed snow height: $h = H := H_{obs}$. Analogously, the layer's snow water equivalent equals total snow water equivalent: $swe = SWE := \rho_0 \cdot H_{obs}$, with fresh snow density $\rho_0$ being an important parameter of the $\Delta$SNOW.MODEL (cf. Sect. 3). The treatment of the first snow event is illustrated at $t = 2$ in Fig. 1; it is processed within the Fresh Snow Module (Sect. 2.2.1).

### 2.1 Dry Metamorphism

As it was mentioned in the Introduction, Martinec and Rango (1991) used a power function (Eq. (2)) to describe densification of aging snow, because this way errors in initial density $\rho_0$ get less relevant over time. For the $\Delta$SNOW.MODEL this kind of high error tolerance of $\rho_0$ is a rather feeble argument to use a power law, since it only holds for old snow and deep snowpacks, but with the $\Delta$SNOW.MODEL also $SWE$ of ephemeral snowpacks (e.g., at low elevation sites) should be modeled as good as possible. Furthermore, as the $\Delta$SNOW.MODEL considers overburden load in a particular way (Sect. 2.2.1), it is not expedient to have a direct dependence between density and age of a layer. Aside from that drawbacks of Martinec and Rango (1991)'s power law compaction (and in contrast to their unproven claim "snow would [not] settle [. . . ] according to an exponential curve"), most snow models very well simulate snow compaction by way of Newtonian viscosity with associated exponential densification over time (e.g., Jordan et al., 2010). The $\Delta$SNOW.MODEL's Dry Metamorphism Module combines the effects of dry metamorphism and deformation strain, by applying the following adaption of Sturm and Holmgren (1998)'s relation, with





the help of De Michele et al. (2013). (For wet metamorphism – as defined by Jordan et al. (2010) – see Drenching Module in
Sect. 2.2.3.)

$$\frac{h(i,t-1)}{h(i,t)} = 1 + \Delta t \cdot \frac{\widehat{\sigma}(i,t)}{\eta(i,t)}$$

$$\text{with } \widehat{\sigma}(i,t) = g \cdot \sum_{\hat{i}=i}^{ly(t)} swe(\hat{i},t) \tag{3}$$

$$\text{and } \eta(i,t) = \eta_0 \cdot e^{k \cdot \rho(i,t)}.$$

Model timestep $\Delta t$ in general is arbitrary, but usually it is one day. If so, $t$ can be explained as "today" and $t-1$ as "yesterday"
here. Accordingly, $h(i,t)$ is today's modeled height of the $i$-th snow layer. Snow layers are counted from bottom to top; layer
$i = 1$ is the lowest and oldest layer. Today's height of the total snowpack is $H(t) = \sum_i h(i,t)$.

The individual snow water equivalents of the layers are given by $swe(i,t)$, and their sum represents total mass of the
snowpack $SWE(t) = \sum_i swe(i,t)$. The vertical stress at the bottom of layer $i$ is given by $\widehat{\sigma}(i,t)$ (De Michele et al., 2013). It is
constituted by the sum of loads overlying layer $i$ (including layer $i$'s own load), with $ly(t)$ being today's total number of snow
layers or – in other words – $ly(t)$ is the index $i$ of today's uppermost (i.e., "surface") layer.

The Newtonian viscosity of snow $\eta$ is made density-dependent in the framework of the $\Delta$SNOW.MODEL (following Kojima,
1967), but dependencies on temperature, grain characteristics etc. are very consciously ignored – due to the lack of information
on it when dealing with pure snow height data. Today's density of layer $i$ is $\rho(i,t)$; it equals $\frac{swe(i,t)}{h(i,t)}$. $k$ and $\eta_0$ are tuning
parameters of the Dry Metamorphism Module (see Sect. 3).

To avoid excessive compaction a crucial parameter is introduced in the $\Delta$SNOW.MODEL: $\rho_{\max}$. It defines the maximal pos-
sible density of a snow layer and (consequently) also the maximum bulk snow density. Finding its optimal value is subject to
calibration (Sect. 2.3). $\rho_{\max}$ figures the density a snow layer (or the whole snowpack) can reach at most, unless it looses mass
by melting. $\rho_{\max}$, of course, is a model parameter and cannot be observed in real snowpacks. In case the Dry Metamorphism
Module increases the density of one or more layers beyond $\rho_{\max}$, $\rho(i,t)$ of the respective layer(s) is set equal to $\rho_{\max}$.

According to Eq. (3) the rate of densification of a certain snow layer is linearly depending on the overlying snow load $\widehat{\sigma}(i,t)$
and exponentially depending on the layer's density $\rho(i,t)$. Sturm and Holmgren (1998) conclude that this difference is one
reason why "snow load plays a more limited role in determining the compaction behavior than grain and bond characteristics
and temperature". Nonetheless, higher overloads result in stronger compaction. Denser and older layers, respectively, compact
less than newer layers with lower densities. This links the densification rate to the layer age, but indirectly by the use of density.
Therefore, $\Delta$SNOW.MODEL's compaction is not directly depending on layer age as it was the case if using Martinec and Rango
(1991)'s power law. The usage of an exponential function for compaction is one major difference between the $\Delta$SNOW.MODEL
and Gruber (2014), who uses a power law approach similar to Martinec and Rango (1991).

The Dry Metamorphism Module of the $\Delta$SNOW.MODEL is illustrated by the light blue arrows in Fig. 1. This module is
applied at every point in time (except if there is no snow; see $t = 1$ in Fig. 1). The Dry Metamorphism Module is the "highest-





ranking" module because based on its result the $\Delta$SNOW.MODEL decides between three different processes, realized by the other three modules:

## 2.2  Process Decisions

The $\Delta$SNOW.MODEL simulates the layers' heights for the next point in time $h(i,t)$ using Eq. (3) (Dry Metamorphism Module). As one time step typically equals one day, figuratively spoken the Dry Metamorphism Module acts "over night", from "yesterday" to "today". "Today" observed $H_{\mathrm{obs}}(t)$ and modeled $H(t)$ are compared. The $\Delta$SNOW.MODEL's process decision algorithm now takes the result of the difference $\Delta H(t) = H_{\mathrm{obs}}(t) - H(t)$ and confronts it with $\tau$ [m]. $\tau$ is another tuning parameter of the $\Delta$SNOW.MODEL (see Sect. 2.3). Its value is in the order of a few centimeters (see Sect. 3) since $\tau$ could also be regarded as a measure for observational error. Technically, $\tau$ defines a limit of $\Delta H(t)$ whose overshooting, adherence, and undershooting heads for one out of three branches, which – together with the Dry Metamorphism Module – build the four modules of the $\Delta$SNOW.MODEL:

- Fresh Snow (2.2.1): In case observed snow height is significantly higher than modeled snow height ($\Delta H(t) > +\tau$), a snowfall event is assumed. This means mass gain as well as enhanced compaction of underlying layers due to the overburden load.

- Scaling (2.2.2): In case there is no significant difference between observed and modeled snow height ($-\tau \leq \Delta H(t) \leq +\tau$), neither mass gain nor loss is modeled. However, modeled snowpack is "scaled", i.e. compressed or stretched, to fulfill the condition $H(t) = H_{\mathrm{obs}}(t)$.

- Drenching (2.2.3): In case observed snow height is significantly lower than modeled snow height ($\Delta H(t) < -\tau$), it is interpreted as wet snow metamorphism. In the snowpack this "drenching" happens from top to bottom, resulting in the associated (strong) decline of snow height. The drenching can either be caused by melt (mass loss) or rain (mass gain), whereas treating the latter is optional, not finalized in the current version of the $\Delta$SNOW.MODEL, and not further detailed here.

### 2.2.1  Fresh Snow

In case $\Delta H(t) > +\tau$, meaning observed snow height is significantly higher than modeled snow height, a fresh snow event is supposed and a new top snow layer is modeled by the $\Delta$SNOW.MODEL (see at $t = 2$ and $t = 7$ in Fig. 1 for a schematic illustration). This is a consequential step and nothing innovative at all. Other models have implemented this mechanism as well (e.g., Martinec and Rango, 1991; Sturm et al., 2010). However, the $\Delta$SNOW.MODEL goes beyond and introduces another feature: It explicitly models the peculiar effect of overburden load on underlying layers, defined as their enhanced densification due to (sudden) stress, which is put on by the weight of fresh snow (or rain-on-snow). Grain bonds get broken, grains slide, partially melt, and warp (Jordan et al., 2010), and the layers densify comparatively rapidly and strongly. The $\Delta$SNOW.MODEL interprets overburden load as an "unsteady and discontinuous" stress on the snowpack, under which snow presumably does not





react as a viscous Newtonian fluid. (As long as $\Delta t$ – the time between two consecutive observations – is in the order of at least some hours, discontinuity is an intrinsic feature of the process. Mostly daily snow heights are available, a fresh snow event is always an unsteady case then.) The part of Jordan et al. (2010)'s deformation strain, that is created by the individual layer loads, is interpreted as a "continuous" effect and is processed by the Dry Metamorphism Module; see $\widehat{\sigma}$ in Eq. (3). The Fresh Snow Module realizes the effect of overburden load by reducing each layer's height $h(i,t)$ with the help of the dimensionless "overburden strain" $\epsilon(i,t)$, defined as

$$\epsilon(i,t) = c_{\mathrm{ov}} \cdot \sigma_0 \cdot e^{-k_{\mathrm{ov}} \frac{\rho(i,t)}{\rho_{\max} - \rho(i,t)}}$$

$$\text{with } \sigma_0 = \Delta H(t) \cdot \rho_0 \cdot g.$$

(4)

$c_{\mathrm{ov}}$ [Pa$^{-1}$] is another tuning parameter of the model (see Sect. 2.3). It controls the importance of the sudden, enhanced compaction due to overburden load. According to Sturm and Holmgren (1998) and in consistency with Eq. (3) snow load has a linear effect on the bulk density. Therefore, $\epsilon(i,t)$ is made linearly depending on the load, which the overlying fresh snow is putting on the underlying layers. This load (or stress or pressure) is well approximated by $\sigma_0$ [Pa]; the bigger the overburden load, the stronger the compaction. (The overburden load does not fully equal $\sigma_0$, since $\Delta H(t)$ is not the height of the fresh snow, but the difference between modeled height – "before" knowing about the fresh snow event – and observed height "after" the fresh snow event. An iterative calculation would be more precise, however, Eq. (4) proved to be an adequate compromise between simplicity and accuracy.) In order to avoid $\epsilon(i,t) > 1$, $c_{\mathrm{ov}}$ is restricted (at least) to the range of values between 0 and the minimum value of the data record for $\frac{1}{\sigma_0}$. As $\sigma_0$ hardly ever exceeds 1000 Pa, $\frac{1}{\sigma_0}$ normally is larger than $1 \times 10^{-3}$ Pa$^{-1}$. This value, thus, marks a good upper bound for $c_{\mathrm{ov}}$ (Sect. 2.3). Dimensionless $k_{\mathrm{ov}}$ controls the role of a certain snow layer's density (i.e., age, respectively) on $\epsilon(i,t)$, and has to be specified by calibration (see Sect. 2.3). The density-dependence of $\epsilon(i,t)$ was chosen to be exponential, and using $\rho_{\max}$ in the denominator of Eq. (4)'s exponent secures that overburden loads cannot make snow layers denser than $\rho_{\max}$. The closer a snow layer's density is to the maximum density $\rho_{\max}$, the less it will be compacted by additional load. Relatively new and – therefore – not very dense layers, are exposed to greater densification, which is exactly what is observed in reality. As it will be shown in sections 2.2.2 and 2.2.3 $\rho_{\max}$ also governs mass loss and melt in the model. Not least, $\rho_{\max}$ illustrates the possible maximum density of a wet seasonal snowpack in the $\Delta$SNOW.MODEL world and – as it can be seen in Sect. 3 – it is possible to assign a reasonable value to it. (Sturm et al. (2010), who revisited Sturm and Holmgren (1998), already introduced a maximum density for seasonal snow. They used it very prominently in their formula for modeling bulk density and defined five (snow) climate classes with different values of $\rho_{\max}$ ranging from 217 to 598 kg m$^{-3}$.)

The "overburden strain" $\epsilon(i,t)$ theoretically lies between 0 and 1 and compresses all (old) snow layers of the model in case of a fresh snow event. Practically, $\epsilon(i,t)$ is often close to zero (in this study 90% of all computed $\epsilon$ are smaller than 0.09) and extremely rarely higher than 0.3 (in this study only 9 out of 10000).





The following intermediate (asterisked) variables are defined due to the overburden load. The compressed layer's masses, $swe(i,t)$, remain unaffected during this process.

$$\epsilon(i,t) = \frac{h(i,t) - h^*(i,t)}{h(i,t)} \quad \text{leading to} \quad h^*(i,t) = (1 - \epsilon(i,t)) \cdot h(i,t)$$

$$H^*(t) = \sum_i h^*(i,t) \tag{5}$$

$$\rho^*(i,t) = \frac{swe(i,t)}{h^*(i,t)}$$

A fresh snow event, identified by the condition $\Delta H(t) > +\tau$, of course not only impacts the older snow and compacts it more strongly, but it also adds a new snow layer and mass to the snowpack (pink arrow at $t = 2$ and $t = 7$ in Fig. 1). The number of layers is increased by one and the following attributes are given to the new layer:

$$age(ly,t) = 1$$

$$h(ly,t) = H_{obs}(t) - H^*(t) \tag{6}$$

$$swe(ly,t) = h(ly,t) \cdot \rho_0$$

The total snow water equivalent is risen: $SWE(i,t) = SWE(i,t-1) + swe(ly,t)$, and the intermediate variables of Eq. (5) overwrite their originals: $h(i,t) = h^*(i,t)$, $H(t) = H^*(t) + h(ly,t)$, and $\rho(i,t) = \rho^*(i,t)$. The model snowpack with this new properties now again compacts according to Eq. (3), time $t$ is risen by one increment, and "tomorrow" the process again starts with the decision described in Sect. 2.2.

The treatment of overburden snow loads as triggers for enhanced compaction of the underlying snow is illustrated with a purple arrow at $t = 7$ in Fig. 1. Although the overburden strain $\epsilon$ in most cases hardly deviates from zero (see above), the value of this feature for the performance of the $\Delta$SNOW.MODEL is supposed to be rather high, at least it should be worth the effort (cf. Sect. 5).

### 2.2.2 Scaling

Of course, equations (3 and (4 are highly simplified representations of the complex viscoelastic behavior of snow and make no claims of being particularly precise. Still, also snow height observations typically only show an accuracy of a few centimeters. The $\Delta$SNOW.MODEL accepts these inherent inaccuracies and apparent discrepancies between model and measurements and copes with them by not applying too strict criteria in the process decisions described in Sect. 2.2. The uncertainty measure $\tau$ (introduced above) acts as a buffer to avoid too frequent gain or loss of mass in the model world: In case $|\Delta H| \leq |\tau|$ neither the snowpack looses mass nor gains mass, but mass is kept constant. In order to benefit from having a new measurement at every point in time, $H(t)$ is intentionally set to $H_{\text{obs}}(t)$ by the Scaling Module.

The Scaling Module forces a partial reversal of the previous compaction, which was modeled by the Dry Metamorphism Module between $t - 1$ and $t$. The best-fitted parameter setting for $\eta_0$ is temporarily rejected and substituted by $\eta_0^*$. It would be





straight forward to use one adjusted $\eta_0^*(t)$ for all layers. However, this leads to a rational function with multiple solutions for
$\eta_0^*(t)$. Consequently, this approach shows a clear non-physical behavior making it necessary to calculate different $\eta_0^*(i,t)$ for
each layer $i$. See Appendix A for details on that.

$\eta_0^*(i,t)$ is then used instead of $\eta_0$ in Eq. (3) to recalculate the compaction of individual layers. $H(t)$ now equals $H_{\mathrm{obs}}(t)$. In
most cases all layers get "slightly more" or "slightly less" compacted by the Scaling Module than by the Dry Metamorphism
Module. Only at rare occasions the scaling does not compact, but a small "stretching" of the snowpack is necessary. This
only happens if there was a small increase in observed snow height *and* very little modeled dry metamorphic compaction; the
condition $H(t) + \tau > H_{\mathrm{obs}}(t) > H_{\mathrm{obs}}(t-1)$ has to be fulfilled. Of course, such "stretching" does not occur in reality, but also
in the $\Delta$SNOW.MODEL it is an infrequent case that only acts at a small scale: in any case the "stretching" is smaller than $\tau$. The
issue is accepted as a model artifact, not least, because the "stretching" enables the very valuable adjustment to $H_{\mathrm{obs}}$ at every
point in time – without forcing mass gain for any insignificant $H$ raises that are within the measurement accuracy.

In case the density of an individual layer exceeds $\rho_{\max}$ by the scaling process, the excess mass is distributed layerwise from
top to bottom. $SWE$ remains constant during scaling, unless it would be necessary to compact all layers beyond $\rho_{\max}$. In this
case the appropriate excess mass is taken from the model snowpack and interpreted as runoff, $SWE$ is reduced and all layer
heights are cut accordingly (see Sect. 2.2.3 for details). As $\tau$ turns out to be – reasonably and preferably – chosen in the order
of a few centimeters (Sect. 3), the resulting reduction of $SWE$ within the Scaling Module is always quite small (e.g., with
$\tau = 2\,\mathrm{cm}$ and maximum density is $450\,\mathrm{kg\,m}^{-3}$ the mass loss, i.e. runoff, is only $9\,\mathrm{kg\,m}^{-2}$).

The Scaling Module is illustrated as black arrows in Fig. 1. Note again that the scaling is nothing "physical", but also nothing
"substantial" in terms of $SWE$, yet it is a smart way to utilize the advantage of having a measured snow height at every point
in time.

### 2.2.3   Drenching

The Drenching Module, finally, defines compaction due to liquid water percolating from top to bottom through the snowpack,
loosening grain bonds and leading to densification. In case observed snow height at a certain point in time is significantly lower
than modeled snow height ($\Delta H(t) < -\tau$), the Drenching Module is activated. Drenching compaction is the $\Delta$SNOW.MODEL's
synonym of wet snow metamorphism.

The drenching can either be caused by melt or rain and the $\Delta$SNOW.MODEL can principally deal with both processes, which
are (often) contradictory in terms of mass change (melt: mass loss or invariant; rain: mass gain or – only if combined with runoff
– invariant or mass loss). However, distinguishing between them is indeed extremely difficult (if not impossible) if only snow
heights are available. For the time being the $\Delta$SNOW.MODEL ignores rain since it concentrates on modeling $SWE$ for pure
snow height records without having any further information on e.g precipitation, temperature, snowfall level,. . . Possibilities
how rain could be addressed at future developments are outlined in Sect. 5. This drawback seems disappointing, however, given
the relative success of the $\Delta$SNOW.MODEL "without rain" (see Sect. 3) one should not expect too much improvement when
incorporating rain in one or another – potentially elaborate – way.



To cope with the model/observation discrepancy $\Delta H(t) < -\tau$ the Drenching Module densifies the model layers until $\rho_{\max}$ is reached – starting from the uppermost one. Figuratively spoken, it wettens or drenches a certain layer until it is "saturated" and further distributes the melt water to the underlying layer. This process is repeated until (transient, therefore asterisked) $H^*$ equals $H_{\mathrm{obs}}(t)$. One or more (upper) layers might reach $\rho_{\max}$. In case $\Delta H(t)$ is so negative that all model snow layers (from

top to bottom) are compacted and densified to $\rho_{\max}$, but still $H^* > H_{\mathrm{obs}}(t)$, the product of the remaining height difference and the maximum density constitutes mass loss, i.e. runoff $R(t)$:

$$R(t) = (H^* - H_{\mathrm{obs}}(t)) \cdot \rho_{\max}. \tag{7}$$

All layer heights are "cut" by a respective portion: $(H^* - H_{\mathrm{obs}}) \cdot \frac{h^*}{H^*}$. This mechanism does not reduce total number of layers, but layers potentially get very thin. During the melt season, where most of the runoff is produced, the Drenching Module is

more or less continuously active until it is snow-free $(H_{\mathrm{obs}}(t) = 0)$ and all the snow has been converted to runoff. For one distinct snowpack – from the first snow fall $(t_1)$ until getting snow-free again $(t_2)$ – one has $\sum_{t_1}^{t_2} R(t) = SWE_{\mathrm{pk}}$.

In Fig. 1 the Drenching Module is shown by the brown arrow (as long as there is no mass loss) and by the cyan arrow (in case runoff is modeled).

## 2.3   Calibration

The $\Delta$SNOW.MODEL has seven parameters that can be used for calibration: $\rho_0$, $\rho_{\max}$, $\eta_0$, $k$, $\tau$, $c_{\mathrm{ov}}$, and $k_{\mathrm{ov}}$ (cf. Tab 1). For the first four parameters one finds suggestions and ranges in the literature:

Sturm and Holmgren (1998) do not address the criticality for the choice of fresh snow density, however, they use constant $\rho_0 = 75\,\mathrm{kg\,m^{-3}}$. It is a well known characteristic of freshly fallen snow to show large variations in densities. Helfricht et al. (2018) reviewed many studies and give a general range of $10 - 350\,\mathrm{kg\,m^{-3}}$, narrowing it down to "mean values" be-

tween $70 - 110\,\mathrm{kg\,m^{-3}}$. Note, that this is daily densities. Sub-daily means of fresh snow densities are lower. Helfricht et al. (2018), for example, come up with an average of $68\,\mathrm{kg\,m^{-3}}$ for hourly time intervals. During the calibration process for the $\Delta$SNOW.MODEL $\rho_0$ was varied from 50 to $200\,\mathrm{kg\,m^{-3}}$.

The second density-related calibration parameter is $\rho_{\max}$, the maximum possible density within the model framework. As mentioned, Sturm et al. (2010) already defined such a maximum for five different climate classes. They range from

217 to $598\,\mathrm{kg\,m^{-3}}$. Glaciologists set the "critical density" before snow turns into firn (which is wetted snow that has survived one summer) to 400 to $800\,\mathrm{kg\,m^{-3}}$ (e.g., Paterson, 1998). Still, manual density measurements of seasonal snow used in previous studies hardly ever exceeded $\rho_b = 500\,\mathrm{kg\,m^{-3}}$ (e.g., Jonas et al., 2009; Guyennon et al., 2019). Armstrong and Brun (2010) limit it to approximately 400 to $500\,\mathrm{kg\,m^{-3}}$ too. In order to find the fittest value for $\rho_{\max}$ used in the $\Delta$SNOW.MODEL, it was varied from 300 to $600\,\mathrm{kg\,m^{-3}}$.

Equation (3) needs $\eta_0$, the "viscosity at $\rho$ equals zero" (Sturm and Holmgren, 1998). It is found to be in the order of $8.5 \times 10^6\,\mathrm{Pa\,s}$ (Sturm and Holmgren, 1998), $6 \times 10^6\,\mathrm{Pa\,s}$ (Jordan et al., 2010), and $7.62237 \times 10^6\,\mathrm{Pa\,s}$ (Vionnet et al., 2012). During the calibration process for the $\Delta$SNOW.MODEL $\eta_0$ was varied from 1 to $20 \times 10^6\,\mathrm{Pa\,s}$. Parameter $k$, the second neces-





sary parameter in Eq. (3), was varied from $0.011$ to $0.08\,\mathrm{m^3\,kg^{-1}}$ by Sturm and Holmgren (1998) depending on climate region and respective different types of snow. However, they cite Keeler (1969) in their Table 2 with values for $k$ for "Alpine-new"
snow of up to $0.185\,\mathrm{m^3\,kg^{-1}}$. In more complex snow models $k$ is set to $0.023\,\mathrm{m^3\,kg^{-1}}$ (see Crocus: $b_\eta$ in Vionnet et al. (2012)'s Equation (7); and also in Equation (2.11) of Jordan et al., 2010) or $0.021\,\mathrm{m^3\,kg^{-1}}$ (see SNTHERM: Equation (29) in Jordan, 1991). Its range for the $\Delta$SNOW.MODEL calibration was set from $0.01$ to $0.2\,\mathrm{m^3\,kg^{-1}}$, which is quite generous.

There are no references for the latter three parameters. $\tau$, as mentioned, might be interpreted as a measure of observation error, is regarded to be in the order of a few centimeters, and was modified from $1\,\mathrm{cm}$ to $20\,\mathrm{cm}$ for calibration. The last two
parameters determine the role of overburden strain and are kind of a specialty of the $\Delta$SNOW.MODEL: $c_{\mathrm{ov}}$ and $k_{\mathrm{ov}}$. At least the limits of $c_{\mathrm{ov}}$ could be defined (Sect. 2.2.1) as $c_{\mathrm{ov}} \in \left[0, \min(\frac{1}{\sigma_0})\right]$. $k_{\mathrm{ov}}$ is only known to be a (dimensionless) real, positive number. For calibrating the $\Delta$SNOW.MODEL $c_{\mathrm{ov}}$ and $k_{\mathrm{ov}}$ were restrained by $[0, 10^{-3}\,\mathrm{Pa^{-1}}]$ and $[0.01, 10]$.

As mentioned, timestep $\Delta t$ principally can be chosen arbitrarily. Mostly it might be one day, because many (longterm) snow height measurements are on a daily basis. The calibration performed in this study is based on $\Delta t = 1\,\mathrm{day}$. Still, longer $\Delta t$
(e.g., three days) as well as shorter $\Delta t$ (e.g., one hour) are conceivable and could be handled by the $\Delta$SNOW.MODEL. Note, however, (at least some) calibration parameters will change significantly when changing $\Delta t$. This gets obvious when thinking about fresh snow density $\rho_0$, which of course is different if defined for one hour or for a three day timestep. The usage of this publication's calibration parameters can, therefore, only be suggested for daily snow height records.

### 2.3.1 Calibration Data and Method

The calibration process needs data – either from observations or from a much more sophisticated snow model, whose simulated $SWE$s are sufficiently reliable.

As it was outlined in the Introduction, $SWE$ measurements are quite rare. However, for calibration not only $SWE$ observations are needed, but also gapless snow heights records from the same places, at least at daily resolution. Gruber (2014) collected 15 years of weekly $SWE$ data from six stations in the Eastern Alps, measured by the observers of the Hydrographic
Service of Tyrol (Austria) between winters 1998/99 and 2012/13. The measurements of snow height and water equivalent were made manually in snow pits with rulers and snow sampling cylinders ($500\,\mathrm{cm^3}$), respectively. The sites range from $590\,\mathrm{m}$ to $1650\,\mathrm{m}$ altitude and are situated in relatively dry, inneralpine regions as well as in the Northern and Southern Alps, which are more humid due to orographic enhancement of precipitation (see Gruber, 2014, for details). The sites in the Southern Alps even show a moderate maritime influence due to their vicinity to the Mediterranean Sea, the most important source of moisture
for this region (e.g., Seibert et al., 2007). These $6 \times 15 = 90$ winter seasons cover 1166 measured $H$–$SWE$ pairs. Besides these $SWE$ measurements manual $H$ measurements are available for every day at the respective stations.

The second source for $SWE$ measurements used for calibration is Marty (2017). The Swiss SLF freely provides biweekly $SWE$ and daily $H$ data from 11 stations in Switzerland (mostly in the Northern Alps, some inneralpine) spanning an altitude range from $1200\,\mathrm{m}$ to $2540\,\mathrm{m}$. The biweekly $H$ measurements (corresponding to the $SWE$ measurements) were compared
with the daily $H$ records. Only those sites and years were used for calibration where the respective values of the daily $H$ record match the values of the biweekly measurements. This was the case for 9 stations, with all in all 56 winters and 363 pairs



of $H$ and $SWE$ measurements. (Other stations and years suffer from discrepancies caused by too far distances between the measurements etc.)

In order to ensure an unperturbed validation, the observation data sets from Austria and Switzerland (1529 $SWE - H$ pairs)

were split into even years for model calibration ($SWE_{\mathrm{cal}}$) and odd years for validation ($SWE_{\mathrm{val}}$).

Model calibration was performed with the statistical software *R* (R Core Team, 2019) and the *R* package *optimx* (Nash, 2014). Results were obtained with optimization methods *L-BFGS-B* followed by *bobyqa* which both are able to handle lower and upper bounds constraints. The function to be minimized was the root mean squared error of $SWE$s from the $\Delta$SNOW.MODEL and observed $SWE$s, using the calibration data set $SWE_{\mathrm{cal}}$.

## 400  3   Results

The following evaluates the ability of the $\Delta$SNOW.MODEL to calculate snow water equivalents exclusively from snow heights and its practicability. See sections 3.2 and 4 for more. Before that, however, the model parameters have to be optimized during the calibration process (which was described in Sect. 2.3). The results thereof – the best-fitted parameters – are important model results per se, since they show insights how well model world and real world fit together.

### 405  3.1   Optimized Parameters and Sensitivities

Table 1 gives an overview of all parameters and summarizes the optimal setting for the $\Delta$SNOW.MODEL. A graphical analysis of the model sensitivity to parameter changes is shown in Fig. 2. $SWE_{\mathrm{pk}}$ was chosen to indicate sensitivity, since it is probably the most important quantity for snow-hydrology and other fields where snow mass is a key variable (see Introduction).

#### 3.1.1   Fresh Snow Density $\rho_0$

Being aware of both – the huge possible variations of fresh snow density $\rho_0$ depending on meteorological conditions during snowfalls and the possible cruciality of this parameter for $SWE$ simulation by the $\Delta$SNOW.MODEL – $\rho_0$ was chosen to be a constant in the framework of the model. For $\rho_0 = 81\,\mathrm{kg\,m^{-3}}$ the minimal root mean square differences/errors (RMSE) between all $SWE$ observations used for calibration ($SWE_{\mathrm{cal}}$) and the respective modeled values was reached. This value clearly lies within the broader frame of possible fresh snow densities and quite closely to Sturm and Holmgren (1998)'s $75\,\mathrm{kg\,m^{-3}}$ (The

$\Delta$SNOW.MODEL could be seen as an extended combination of the Sturm and Holmgren (1998) and Martinec and Rango (1991) approaches.), but it is found in the lower part for "typical" fresh snow densities (e.g., Helfricht et al., 2018). A possible explanation could be that the $SWE$ measurement records used for the calibration tend to underrepresent late winter and spring conditions. Regular (weekly, biweekly) observations capture the short melt seasons worse than the (much) longer accumulation phases. Therefore, $SWE$ records might be biased towards early and mid winter fresh snow densities, which are lower (e.g.,

Jonas et al., 2009). Still, there are also some indications that using e.g., $100\,\mathrm{kg\,m^{-3}}$ as constant fresh snow density when modeling $SWE$ results in an overestimation of precipitation (up to 30% according to Mair et al., 2016). The calibrated value



for $\rho_0$ can be regarded as a reasonable result, even more when only considering it as a model parameter but not as a physical constant.

The sensitivity analysis illustrated in Fig. 2 confirms the importance of a good choice of $\rho_0$. Increasing $\rho_0$ quite fast leads
to a decrease of the relative bias of seasonal $SWE$ maxima ($SWE_{\mathrm{pk}}$). (Note the definition of the relative bias in Fig. 2's caption.) In absolute values: too small $\rho_0$ cause too small $SWE_{\mathrm{pk}}$, using higher values leads to an overestimation of $SWE_{\mathrm{pk}}$. This behavior supports above-mentioned tendency to overestimate precipitation when choosing constant $100\,\mathrm{kg\,m^{-3}}$ as fresh snow density. As expected, the fresh snow density is the most crucial parameter of the $\Delta$SNOW.MODEL (cf. Tab 1). The median relative bias of $SWE_{\mathrm{pk}}$ changes by -0.46% per $+1\,\mathrm{kg\,m^{-3}}$, if the whole calibration range of $\rho_0$ is considered to calculate the
sensitivity ($50-200\,\mathrm{kg\,m^{-3}}$). This means a median change in $SWE_{\mathrm{pk}}$ of $+0.37\,\mathrm{kg\,m^{-2}}$ when $\rho_0$ is risen by $+1\,\mathrm{kg\,m^{-3}}$. If the limits are chosen tighter around the optimal value, the gradient is even steeper: -0.62% and $+0.50\,\mathrm{kg\,m^{-2}}$ per $+1\,\mathrm{kg\,m^{-3}}$, respectively, when the gradient is approximated for the range $70-90\,\mathrm{kg\,m^{-3}}$. Widely-used $\rho_0 = 100\,\mathrm{kg\,m^{-3}}$, consequently, causes a median overestimation of $SWE_{\mathrm{pk}}$ of about 12% in the $\Delta$SNOW.MODEL (same for daily $SWE$). Users should be aware of this. The suggestion clearly is to either use the best-fitted data of this study or recalibrate *all* parameters (with appropriate
$SWE$ data), but not adjusting only single parameters. As the calibration data of this study are spread across various climates and altitudes, users can be quite confident to get good results if using $\rho_0 = 81\,\mathrm{kg\,m^{-3}}$. This value seems to be a good compromise – at least at alpine areas. However, for (very) maritime, very dry, polar or tundra regions the optimized $\rho_0$ should be used with caution; if possible, recalibration is recommended.

### 3.1.2 Maximum Density $\rho_{\mathrm{max}}$

Of course, the maximum bulk snow density of a snowpack changes from year to year and site to site. For the $\Delta$SNOW.MODEL simplicity and independence from meteorological variables outweigh precision. Even more so, when there are good arguments for the existence of a "typical" maximum bulk density $\rho_{\mathrm{max}}$. Put simply, (not too old) seasonal and also ephemeral snowpacks melt away when they get water saturated. Before that, there is limited time for dry densification; dry winter snow's bulk density is widely described as staying below about $350\,\mathrm{kg\,m^{-3}}$ (e.g., Paterson, 1998; Sandells et al., 2012). Accounting for
the fact that volumetric liquid water content of about 10% marks the funicular mode of liquid distribution in old, coarse-grained snow (Denoth, 1982; Mitterer et al., 2011), this leads to the rough estimate of a typical maximum bulk density of about $\frac{9}{10} \cdot 350 + \frac{1}{10} \cdot 1000 = 415\,\mathrm{kg\,m^{-3}}$. Convincingly, the fittest value for $\rho_{\mathrm{max}}$ in the $\Delta$SNOW.MODEL turns out to be $401\,\mathrm{kg\,m^{-3}}$, which is close to that value and well situated within the range given in the literature (Table 1). Moreover this is virtually the same value like the median maximum seasonal density of the $SWE_{\mathrm{val}}$ data records ($400\,\mathrm{kg\,m^{-3}}$, see box plot in Fig. 3) –
another indication why $\rho_{\mathrm{max}}$ could be regarded as a typical seasonal maximum of $\rho_{\mathrm{b}}$.

Figure 2 illustrates the similarity between $\rho_0$ and $\rho_{\mathrm{max}}$ regarding their influence on $SWE$ simulations. Keeping the other six $\Delta$SNOW.MODEL parameters constant but increasing $\rho_{\mathrm{max}}$ leads to increased $SWE_{\mathrm{pk}}$ and vice versa – just like $\rho_0$. This is not surprising, however reasonable. The $\Delta$SNOW.MODEL is not as sensitive to changes in $\rho_{\mathrm{max}}$ than to changes in $\rho_0$: Raising $\rho_{\mathrm{max}}$ by $+1\,\mathrm{kg\,m^{-3}}$ leads to a mean decrease of the relative bias of $SWE_{\mathrm{pk}}$ of -0.06%, which corresponds to an increase
in absolute $SWE_{\mathrm{pk}}$ of $+0.24\,\mathrm{kg\,m^{-2}}$ per $+1\,\mathrm{kg\,m^{-3}}$. The same argumentation like for $\rho_0$ in Sect. 3.1.1 lets users of the





ΔSNOW.MODEL be quite sure when taking $\rho_{\mathrm{max}} = 401\,\mathrm{kg\,m^{-3}}$, the best-fitted value according to this study's calibration. Be aware that solely changing parameter $\rho_{\mathrm{max}}$ for an application of the ΔSNOW.MODEL elsewhere, without proper recalibration of the other parameters, might lead to significant changes in the results for $SWE$. For the data set of this study, e.g., using $\rho_{\mathrm{max}} = 500\,\mathrm{kg\,m^{-3}}$ (instead of $401\,\mathrm{kg\,m^{-3}}$) would lead to a median overestimation of $SWE_{\mathrm{pk}}$ of about $24\,\mathrm{kg\,m^{-2}}$.

### 460  3.1.3  Viscosity Parameters $\eta_0$ and $k$

Equation (3) represents the settlement and densification function of the ΔSNOW.MODEL. Viscosity of layer $i$ at time $t$ is defined as $\eta(i,t) = \eta_0 \cdot e^{k \cdot \rho(i,t)}$. Two parameters $\eta_0$ and $k$ act as adjustment screws and have to be calibrated. Literature places $\eta_0$ in the order of some $10^6\,\mathrm{Pa\,s}$, $k$ is supposed to be in the range of $0.01$ to $0.2\,\mathrm{m^3\,kg^{-1}}$ (Table 1). The latter was varied over the mentioned range during the calibration process and – satisfactorily – it's optimized value is $k = 0.0299\,\mathrm{m^3\,kg^{-1}}$, which is 465 very close to the ones used in widely accepted, sophisticated, thermodynamic models (Jordan et al. (2010), Vionnet et al. (2012); see also Sect. 2.3). The range over which $\eta_0$ was varied during the calibration was $1$ to $20 \times 10^6\,\mathrm{Pa\,s}$. Best-fitted value is $8.52 \times 10^6\,\mathrm{Pa\,s}$, pretty close to other studies' values (Table 1).

As far as the ΔSNOW.MODEL's sensitivity to changes in the "viscosity parameters" $\eta_0$ and $k$ is concerned, Fig. 2 shows that an isolated rise of the model snow viscosity – either by enhancing $\eta_0$ or $k$ – increases the relative bias of $SWE_{\mathrm{pk}}$, which means 470 a decrease in absolute values of $SWE_{\mathrm{pk}}$. This behavior is consistent, since higher viscosity reduces the densification rate and the model snowpack tendentially stays deeper. Consequently, increases in observed snow height tend to bring less fresh snow in the ΔSNOW.MODEL (Fresh Snow Module). Finally, simulated $SWE_{\mathrm{pk}}$ is reduced when $\eta_0$ or $k$ are increased and vice versa.

### 3.1.4  Discrepancy Parameter $\tau$

The ΔSNOW.MODEL's parameter to cope with uncertainties in snow height is $\tau$. It is supposed to be in the order of a few 475 centimeters (at maximum). In particular, it should avoid excessive production of snow mass in the model through too frequent simulation of fresh snow events (see Sect. 2.2). $\tau$ is kind of a peculiarity of the ΔSNOW.MODEL and therefore no bounds can be found in literature. It was generously accepted to range between $1$ and $20\,\mathrm{cm}$ for calibration and turned out to be optimal at $\tau = 2.36\,\mathrm{cm}$ (Table 1). Given the wide range of possible values, this is very close to what it would be expected to be as a measure for $H_{\mathrm{obs}}$ accuracy.

Model sensitivity to changes in $\tau$ turns out to be quite low for values in the order of a few centimeters, but the influence on simulated $SWE_{\mathrm{pk}}$ is strongly increasing if $\tau$ is chosen greater than about $5\,\mathrm{cm}$ (Fig. 2). This result makes a lot of sense, if $\tau$ is seen as a measure of observation accuracy, because this is very likely to be better than $5\,\mathrm{cm}$. Like changes in $\eta_0$ and $k$, changes in $\tau$ are indirect proportional to changes in $SWE_{\mathrm{pk}}$ – for a closely related reason: The bigger $\tau$ the more often (small) fresh snow events are not counted as such because the Scaling Module is more frequently activated at the cost of the Fresh Snow 485 Module (see Sect. 2.2). Mass gains are tendentially modeled less frequently and, as a consequence, snow water equivalents get smaller.





### 3.1.5 Overburden Parameters $c_{\mathrm{ov}}$ and $k_{\mathrm{ov}}$

Aside $\tau$, there are two more parameters that are peculiar to the $\Delta$SNOW.MODEL. They are needed to simulate unsteady compaction by overburden load of fresh snow (Eq. (4)). Because of their (presumed) uniqueness in the snow model spec-
trum there is no information available on how to choose them (see Sect. 2.3 for more). However, the calibration produces $c_{\mathrm{ov}} = 5.10 \times 10^{-4}\,\mathrm{Pa}^{-1}$ and $k_{\mathrm{ov}} = 0.379$ as fittest values (Table 1).

As outlined in Sect. 2.2.1, the implementation of overburden strain in the $\Delta$SNOW.MODEL is supposed to be an important aspect of the model. Still, the sensitivity of modeled $SWE_{\mathrm{pk}}$ to changes in either $c_{\mathrm{ov}}$ or $k_{\mathrm{ov}}$ – without changing the respective other – are quite minor. (See Fig. 2 for $c_{\mathrm{ov}}$. $k_{\mathrm{ov}}$ is not shown, because it is comparable, but with different sign.) The reason for
this relative insensitivity of the model to changes in $c_{\mathrm{ov}}$ and $k_{\mathrm{ov}}$ could be the contradicting effects of these two "overburden parameters": Higher $c_{\mathrm{ov}}$ push overburden strain $\epsilon$ towards 1.0 (Eq. (4)), which increases the role of overburden snow. $h^*$ and $H^*$ of Eq. (5 are reduced and, consequently, the new layer thickness (and mass) is increased (Eq. (6)). Higher $c_{\mathrm{ov}}$, therefore, lead to higher $SWE$ and $SWE_{\mathrm{pk}}$. For $k_{\mathrm{ov}}$ it is the opposite, higher values of $k_{\mathrm{ov}}$ cause lower $SWE$.

## 3.2 Validation and Illustration

As pointed out, the observations of snow water equivalent where divided in two sets, one for calibration ($SWE_{\mathrm{cal}}$) and one for validation ($SWE_{\mathrm{val}}$). The question arises, how accurate the manual $SWE$ observations are. The two data sources do not address this point (Gruber, 2014; Marty, 2017). In general, this is not easy to be answered since $SWE$ measurements made with snow sampling cylinders mostly are used as references in comparison studies, without addressing *their* accuracy (e.g., Sturm et al., 2010; Dixon and Boon, 2012; Kinar and Pomeroy, 2015; Leppänen et al., 2018a). The majority of $SWE_{\mathrm{cal}}$ and $SWE_{\mathrm{val}}$
comes from the Hydrographic Service of Tyrol, Austria, where snow sampling cylinders ($500\,\mathrm{cm}^3$) are used (Sect. 2.3.1). The repeatability of this kind of measurement is estimated at $\pm 4\%$ for glacier mass balance studies, with filling height having the largest influence on accuracy (R. Prinz, Univ. of Innsbruck, Austria; pers. comm.). According to Leppänen et al. (2018b), who compared various density samplers, the "variability among the replications of each sampler and variability among the samplers [is] both less than 10% [...] However, the uncertainty introduced by using different samplers was higher than the uncertainty
caused by observer error". Roughly interpreting these density(!) measurement "variabilities" as relative observation errors for $SWE$, the results for absolute accuracy would typically spread across the wide range of about $2$ to $50\,\mathrm{kg\,m}^{-2}$.

Table 2 provides an overview of model uncertainties for $SWE$. Vionnet et al. (2012) find a root mean squared error and bias of $39.7\,\mathrm{kg\,m}^{-2}$ and $-17.3\,\mathrm{kg\,m}^{-2}$, respectively, comparing 1722 manual samplings at Col de Porte (Chartreuse Mountains, France) and Crocus. These seem to be quite pessimistic values, since root mean squared differences found by Langlois et al.
(2009) are significantly lower. (However, they base on much fewer data.) Roughly summarized, $SWE$ observations as well as "first-class" snow models' $SWE$ simulations are associated with comparable uncertainties; RMSEs might be favorably approximated in the order of $10$ to $20\,\mathrm{kg\,m}^{-2}$.

In this study no quantitative comparison with thermodynamic snow models was performed, since they need further meteo-rological data and the focus was on data records constrained to snow heights. However, the $\Delta$SNOW.MODEL was thoroughly





evaluated against ERMs. Figure 4 and Table 2 show the results. Even though ERMs do not need meteorological data, it is not
straight forward to calibrate them for new sites and applications. From the vast number of ERMs (cf. Avanzi et al., 2015) the
ones of Pistocchi (2016) and Guyennon et al. (2019) were chosen to be fitted to $SWE_\text{cal}$ (standard: Pi16, Gu19; calibrated:
Pi16cal, Gu19cal). These models are (quite) new and easy to calibrate. Additionally, an approach simply using a constant bulk
snow density at every point in time was calibrated to fit this study's data. Interestingly, $278\,\mathrm{kg\,m^{-3}}$ turned out to be the optimal

value minimizing root mean squared errors of all $SWE$ values ($\rho_{278}$). Moreover, Jonas et al. (2009) and Sturm et al. (2010)
were used for comparison (Jo09, St10). Unfortunately, these powerful models could not be calibrated with $SWE_\text{cal}$, because
this would have needed much more $SWE - H$ pairs than those about 1500 having been processed in this study. Therefore,
Jonas et al. (2009) and Sturm et al. (2010) were used with their standard parameters, but for Jonas et al. (2009) it was dis-
tinguished between regions: Region 6 (Jo09R6) having the highest and Region 7 (Jo09R7) having the lowest "region-specific

offset", respectively. Other contemporary approaches had to be ignored, mostly because of the problematic transferability of
regional parameters (e.g., McCreight and Small, 2014, or Mizukami and Perica, 2008).

  Figure 4's upper left panel indicates the decent performance of the ERMs at the mean when applied to the validation-half of
the data set. The bias of modeled $SWE$ is quite low and tendentially positive, meaning $SWE$ is often slightly overestimated
by the ERMs. (Distinct values are given in Table 2.) The $\Delta\text{SNOW.MODEL}$ – on the contrary – slightly underestimates $SWE$

on average, the median bias is $-4.0\,\mathrm{kg\,m^{-2}}$. The overall good results for the ERMs is not particularly surprising, since they
are dedicated to perform well *on average*. The specially calibrated versions of Pistocchi (2016) and Guyennon et al. (2019)
show a significantly smaller bias than their originals. The model of Jonas et al. (2009) has the smallest (actually a negative)
bias for their "Region 7", encompassing the dry, inneralpine Engadin as well as parts of the Southern Alps and the very
East of Switzerland (Samnaun), which is partly influenced by orographic precipitation from Northwesterly flows. In terms of

heterogeneity in precipitation climate "Region 7" is comparable to the region where the $SWE$ data of this study comes from.
Sturm et al. (2010) assess the bias for their model (with their "alpine" data set) at $+29\,\mathrm{kg\,m^{-2}}$ with a standard deviation of
$57\,\mathrm{kg\,m^{-2}}$, and they outline that "in a test against extensive Canadian data, 90% of the computed $SWE$ values fell within
$\pm80\,\mathrm{kg\,m^{-2}}$ of measured values". This is a much more conservative estimation than the results for this study would suggest.

  The other three indicators illustrated in Fig. 4 and summarized in Table 2 – bias of seasonal snow mass maximum $SWE_\text{pk}$

(upper right panel) and the root mean squared errors of individual $SWE$ (lower left panel) as well as of $SWE_\text{pk}$ (lower right
panel) – signify the better performance of the $\Delta\text{SNOW.MODEL}$ compared to ERMs:

  ERMs are intrinsically tied to snow height (see Sect. 1.2) and are systematically forced to overestimate $SWE_\text{pk}$. Devel-
opers of ERMs are well aware of this, nevertheless, this is a pity since $SWE_\text{pk}$ is probably the most-wanted snowpack fea-
ture in hydrology, climatology, and extreme value analysis. The $\Delta\text{SNOW.MODEL}$ proofs to perform much better here. The

$\Delta\text{SNOW.MODEL}$'s bias of $SWE_\text{pk}$ is very minor, only $+2.3\,\mathrm{kg\,m^{-2}}$ at median. ERMs typically suffer from (far) too high
$SWE_\text{pk}$ simulations. The reasoning was given in the Introduction: "ERMs calibrated for good estimates of mean $SWE$
(necessarily) fail to model $SWE_\text{pk}$ sufficiently well", since they are "overregulated" by the snow height. Moreover, the
$\Delta\text{SNOW.MODEL}$ does not only work well for the bias in $SWE$ and $SWE_\text{pk}$, but also for the bias in the timing of $SWE_\text{pk}$





(not shown in Fig. 4, but in the last column of Table 2). ERMs tend to model $SWE_{\text{pk}}$ some days too early – the date of modeled
$SWE_{\text{pk}}$ is pulled towards the date of highest $H$. The $\Delta$SNOW.MODEL virtually reduces this bias to zero.

Another satisfactory validation result for the $\Delta$SNOW.MODEL is shown in Fig. 4's lower panels. RMSEs for all $SWE$
values are constantly lower than if modeled with ERMs: a median error of $23.9\,\text{kg}\,\text{m}^{-2}$ ($\Delta$SNOW.MODEL) faces median errors
between $26.6$ and $38.1\,\text{kg}\,\text{m}^{-2}$ (ERMs). Calibrating Pistocchi (2016) and Guyennon et al. (2019) results in some improvement,
at least they perform much better than the $\rho_{278}$ approach after the calibration. The model of Jonas et al. (2009) does a decent
job also without recalibration, which is remarkable. Sturm et al. (2010)'s method probably suffers from the handicap of being
calibrated with data from the Rocky Mountains. For this comparison the "alpine" parameters of Sturm et al. (2010) were taken,
however, conditions might differ (too) much from the European Alps.

Not least as a consequence of the mentioned drawbacks of the ERMs when dealing with distinct values, the advantages
of the $\Delta$SNOW.MODEL get most obvious when modeling $SWE_{\text{pk}}$ (Fig. 4, lower right; Table 2, last three columns). The
$\Delta$SNOW.MODEL manages to have a small range of RMSEs, with a median of $23.1\,\text{kg}\,\text{m}^{-2}$ and 75% of the errors staying be-
low $34.4\,\text{kg}\,\text{m}^{-2}$. These values for $SWE_{\text{pk}}$ are very close to those for (daily) $SWE$, which emphasizes the $\Delta$SNOW.MODEL's
ability to model all individual $SWE$s equally well. The evaluated ERMs have doubled to tripled errors in simulated $SWE_{\text{pk}}$
and rather big spreads. Remarkably, the simple $\rho_{278}$ approach is performing relatively well. The Jonas et al. (2009) model – if
suitable adjusted to regional specialties – performs best, still, well beaten by the $\Delta$SNOW.MODEL.

The $\Delta$SNOW.MODEL root mean squared errors – slightly lower than $25\,\text{kg}\,\text{m}^{-2}$, also for *certain values* like $SWE_{\text{pk}}$ – are
higher than the errors of the thermodynamic snow models (roughly estimated at $10$ to $20\,\text{kg}\,\text{m}^{-2}$), which is not surprising given
the latter's demands on input data and computational power. However, the $\Delta$SNOW.MODEL outperforms empirical regression
models. This can be argued on base of the study in hand (especially Fig. 4), but even more when looking at the ERM studies
themselves: Jonas et al. (2009) provide RMSEs between $50.9$ and $53.2\,\text{kg}\,\text{m}^{-2}$ for their standard model, which are quite high
values compared to the findings of the study in hand (ca. $30\,\text{kg}\,\text{m}^{-2}$ for $SWE$, see Table 2). One explanation could be that
Jonas et al. (2009), as well as other ERM studies, rely on very diverse measurements (Still, lots of them!). The $\Delta$SNOW.MODEL
study only consists of data from selected stations with long and regular $SWE$ readings, where also ERMs seem to work better.
Guyennon et al. (2019) summarize their and other studies' validation results using MAE, the mean absolute error. Table 2
provides the overview, and again it gets obvious, that ERMs perform better with this study's data, than with Guyennon et al.
(2019)'s data.

Figure 1 schematically shows the functioning of the $\Delta$SNOW.MODEL. A practical example is now provided in Fig. 5, based
on the optimal calibration parameters found during this study and using the same colors as in Fig. 1. Kössen, the station
shown, is situated in the Northern Alps at $590\,\text{m}$ above sea level. Although it is a low-lying place it is known to be snowy,
which is, firstly, due to intense orographic enhancement of precipitation associated with Northwesterly to Northeasterly flows
in the respective region (Wastl, 2008) and, secondly, comparably frequent inflow of cold continental air masses from Northeast.
Showing Kössen – rather than a high altitude station – should emphasize the versatile usability of the $\Delta$SNOW.MODEL. It is not
only designed for high areas with deep, long-lasting snowpacks, but also for, e.g., valleys with shallow, ephemeral snowpacks.





Winter 2008/09 was chosen because the ΔSNOW.MODEL shows a typical performance in terms of RMSEs etc. in Kössen then (see Table 2, values in brackets) and because some important, model-intrinsic features can be addressed and discussed:

Late November 2008 brought the first, however transient snowpack of the season (Fig. 5). The ΔSNOW.MODEL identifies two days with snow fall (purple markings) and models two respective snow layers, which can be distinguished by the thin black line in Fig. 5. After about a week the snowpack starts to melt, the snow layers reach $\rho_{max}$ very fast (the blue shading gets dark), and finally all the snow was converted to runoff (cyan markings). In the second half of December there were three days with fresh snow, followed by a strong decline in snow height. In the frame of the ΔSNOW.MODEL this $H$ decrease is only possible,

if the layers "get wet" – the Drenching Module is activated (marked in brown). The layers get denser, starting at the top. However, the decrease was "manageable" by only increasing the two uppermost layer densities to $\rho_{max}$ and making the third layer just a bit denser. Not all layers got to $\rho_{max}$ ("saturated") and no runoff was modeled. The ΔSNOW.MODEL conserves the two dense layers until the end of the winter, which can clearly be seen in Fig. 5. (One could interpret the layers as consisting of melt forms or a refrozen crust. However, interpretations like that require caution, because modeling such detailed layer features

is not the intention of the ΔSNOW.MODEL!) During January Fig. 5 shows a phase where nothing special happens. Modeled values and observations agree to a high extent and only the Scaling Module makes small adjustments (white markings). Small "stretching events" can be recognized, e.g. on January 2$^{nd}$ and 3$^{rd}$, where model snow layers are set less dense in order to avoid too frequent mass gains. (This model behavior was thoroughly described in Sect. 2.2.2.) During continuous snowfalls in February the successive darkening of the blue layer shadings illustrates a phase of consequent compaction, which actually lasts

until March, when strong decreases in $H_{obs}$ (and, thus, in $H$) already start to activate the Drenching Module. Still, runoff is not yet produced. Only in the second half of March the whole model snowpack reaches $\rho_{max}$ ("saturation"). The ablation phase is clearly distinguishable and lets the snowpack vanish quite fast until about April 10$^{th}$, 2009.

    The snow height record of Kössen from 2008/09 was also used to compare different ERMs and the ΔSNOW.MODEL to $SWE$ observations (Fig. 6). These measurements (light blue circles) are part of the $SWE_{val}$ sample and were manually made with

snow sampling cylinders; one after the December 2008 snowfall, and another nine on a nearly weekly base between late January and late March 2009. Figure 6 also provides various model results and some respective key values are given in Table 2. Not surprisingly, thus evidently, the ERMs' $SWE$ curves "follow" the snow height curve (black dashed line). The ΔSNOW.MODEL (red line) does not get the first four measurements decently correct, the ERMs perform better in this illustrative case. (By the way, the "jumpy behavior" of the model of Jonas et al. (2009), criticized by Pistocchi (2016), does not play a role.) But, after

the stronger snowfalls of February, the picture changes indisputably in favor of the ΔSNOW.MODEL. This is a typical pattern, nothing special for Kössen 2008/09 (albeit it is quite pronounced in this illustrative example): The ERMs are too strongly tied to snow height and, therefore, mostly (1) overestimate $SWE_{pk}$, (2) model its occurrence too early, and (3) – most important – force modeled $SWE$ to reduce during pure compaction phases after snowfalls. All these points were discussed in detail earlier, Fig. 6 visualizes them. At the same time, the ΔSNOW.MODEL does a good job in modeling mean and maximum $SWE$, not

only in Kössen 2008/09 but also on average (cf. Fig. 4 and Table 2). Evidently, the ability of the ΔSNOW.MODEL to "conserve" mass during the phases with dry metamorphism is its strongest point.





## 4 Example of Application: Snow Load Map of Austria

In this section an example is given how the $\Delta$SNOW.MODEL can be used to attain a map of snow loads in Austria; snow load $SL = SWE \cdot g$. European Standards (e.g., European Committee for Standardization, 2015) define the "characteristic snow load"
$s_k$ as the weight of snow on the ground with an annual probability of exceedance of 0.02, i.e. a snow load that – on average – is exceeded only once within 50 years. Unfortunately, $SWE$ is not measured on a regular basis at a reasonable number of sites in Austria (and most other countries). The $\Delta$SNOW.MODEL, however, can provide longterm Austrian $SWE$ series from widely available $H$ series, which can in turn be used for a spatial extreme value model. No other snow model is capable of this in a comparable manner, since either $SWE_{\mathrm{pk}}$ is poorly modeled (ERMs) or more meteorological input would be needed
(thermodynamic models). Among several possibilities to spatially model snow height extremes like max-stable processes (see e.g. Blanchet and Davison, 2011), the *smooth modeling* approach of Blanchet and Lehning (2010) can be used when marginals instead of spatial extremal dependence is in focus.

### 4.1 Smooth Modeling

Extremes following a generalized extreme value distribution (GEV; Coles, 2001) with parameters $\mu$, $\sigma$ and $\xi$ can be modeled
in space by considering linear relations for the three parameters of the form

$$\eta(x) = \alpha_0 + \sum_{k=1}^{m} \alpha_k y_k(x) \tag{8}$$

at location $x$, where $\eta$ denotes one of the GEV parameters, $y_1, \ldots, y_m$ are the considered covariates as smooth functions of the location, and $\alpha_0, \ldots, \alpha_m \in \mathbb{R}$ are the coefficients. Assuming spatially independent stations, the log-likelihood function then reads as

$$l = \sum_{k=1}^{K} \ell_k \left( \mu(x_k), \sigma(x_k), \xi(x_k) \right), \tag{9}$$

where $l$ only depends on the coefficients of the linear models for the GEV parameters. This approach was termed *smooth modeling* by Blanchet and Lehning (2010). A smooth spatial model for extreme snow heights in Austria was already presented in Schellander and Hell (2018), using longitude, latitude, altitude, and mean snow height at 421 stations. Considering the strong correlation between snow height and snow water equivalent, it would be natural to spatially model $SWE$ extremes in the same
manner.

### 4.2 Fitting a Spatial Extreme Value Model

For this application 214 stations with gapless snow height observations in and tightly around Austria of the National Weather Service (ZAMG) and the Hydrological Services are used. The dataset has undergone quality control by the maintaining institu-





tions and covers altitudes between $118$ and $2290\,\mathrm{m\,a.s.l.}$ The records have lengths of 43 years and cover winters from 1970/71
to 2011/2012.

In a first step the $\Delta$SNOW.MODEL was applied to these snow height series to achieve 214 data series of $SWE$ across
Austria. (This indeed is the great strength of the $\Delta$SNOW.MODEL and can hardly be done with other methods!) Then the linear
models for the three GEV parameters according to Sect. 4.1 were defined via a model selection procedure. For that purpose a
generalized linear regression was performed between the parameters and the covariates longitude, latitude, altitude, and mean
snow height, which were added in a stepwise manner. Using the Akaike information criterion (AIC; Akaike, 1974), the best
linear model between a given full model ($\mu \sim$ all covariates) and a null model ($\mu \sim 1$) with the smallest AIC was selected.
Using these models and the covariates of the 214 stations, a smooth spatial model for the yearly maxima of the $SWE$ values
was fitted.

## 4.3 Return Level Map of 50-year Snow Load in Austria

The spatial extreme value model developed in the previous section was applied to the SNOWGRID climate analysis (Olefs
et al., 2013) with yearly mean snow heights from 1961 to 2016. The grid features a horizontal resolution of $1 \times 1\,\mathrm{km}$. Some
minor SNOWGRID pixels have unrealistically large mean snow height values, arising from a poor implementation of lateral
snow redistribution at high altitudes (18 pixels, i.e. 0.02% with values between $5$ and $65\,\mathrm{m}$). They are masked for the calculation
of $SWE$ return level maps. The return level map for a return period of 50 years can be seen in Fig. 7.

As expected, due to the strong correlation of the $SWE$ maxima with mean snow height, the largest snow loads are located
in the mountainous areas of Austria. Although the unrealistic mean snow height values of SNOWGRID are masked, the model
produces a number of 59 (0.06%) snow load values larger than $25\,\mathrm{kN\,m^{-2}}$ in a height range between $1500$ and $3700\,\mathrm{m\,a.s.l.}$.
For a model that would be seriously used e.g. in general risk assessment or structural design, this problem could possibly be
tackled with a non-linear relation between $SWE$ maxima and mean snow height or altitude. This is, however, beyond the scope
of this study. Note, that in the actual Austrian standard (Austrian Standards Institute, 2018) there are no normative snow load
values defined above $1500\,\mathrm{m}$ altitude.

All but two locations of the Austrian $SWE$ measurement series that were used for calibration *and* validation of the $\Delta$SNOW.MODEL
(see Sect. 2.3.1) are included in the dataset used to fit the spatial model in Sect. 4.2. Those two stations, Holzgau and Felber-
tauern with 14 years of $SWE$ observations each, are used to qualitatively compare (1) the spatial model fitted in Sect. 4.2,
(2) $SWE$ extremes modeled from daily snow heights with the $\Delta$SNOW.MODEL, and (3) extremes computed "directly" from
(ca. weekly) observed $SWE$ values. Figure 8 gives an idea of the model performance at stations Holzgau and Felbertauern
(see Fig. 7 for their locations). For the lower-lying station Holzgau ($1100\,\mathrm{m\,a.s.l.}$) all three variants overlap very well. The
50-year return level is $4.65\,\mathrm{kN\,m^{-2}}$ for the smooth spatial model, $4.72\,\mathrm{kN\,m^{-2}}$ for the $\Delta$SNOW.MODEL, and $4.8\,\mathrm{kN\,m^{-2}}$ for
the observations. Note, that the latter stem from weekly observations and, therefore, not necessarily reflect the true yearly
maxima, which naturally must be equal or slightly higher. By the way, the corresponding value of $s_k$ from the Austrian snow
load standard for Holzgau is $6.3\,\mathrm{kN\,m^{-2}}$ (Austrian Standards Institute (2018); accessible online at eHORA (2006)).





For the higher station Felbertauern ($1650\,\mathrm{m\,a.s.l.}$) the agreement between $SWE$ from the $\Delta$SNOW.MODEL and observed values is again very good. However, their GEV fits differ significantly. While the fit to the observations shows a negative shape parameter of $\xi = -0.1$, the fit to the values modeled with the $\Delta$SNOW.MODEL gives a positive shape parameter of $\xi = 0.1$,
leading to much larger return levels for higher recurrence times due to the Fréchet-like distribution. It should be pointed out that the GEV fits based on $\Delta$SNOW.MODEL simulations and observations are unreliable, given the short data sample of only 14 yearly maxima. Indeed, by using a sample size of 43 years and borrowing strength from neighboring stations, the spatial model provides the best fit to observations as well as modeled $SWE$ values. The 50-year snow load return values are $6.4\,\mathrm{kN\,m^{-2}}$ for the spatial model, $6.8\,\mathrm{kN\,m^{-2}}$ for the $\Delta$SNOW.MODEL, and $5.7\,\mathrm{kN\,m^{-2}}$ for the fit to the observations. No normative value is
defined for Felbertauern because it is situated higher than $1500\,\mathrm{m\,a.s.l.}$ (Austrian Standards Institute, 2018).

## 5  Discussion and Outlook

Some questions remain. The treatment of rain-on-snow events surely is one of them. The $\Delta$SNOW.MODEL can – in principle – deal with it. Unsteady compaction due to overburden load, for example, is not restricted to fresh snow. It could also be triggered by the mass of rain water – in nature, but also in the framework of the $\Delta$SNOW.MODEL. Still, the respective coding
is not finalized at the moment, because identifying criteria for rain-on-snow events based on pure snow height records is a problem, and its resolving (if at all possible) is beyond the scope of this paper. In case some meta information on, e.g., rain climate (maybe for a stochastic "rain generator") or on precipitation (no matter if liquid or solid) combined with information on the snowfall line is available, it could quite easily be incorporated in the $\Delta$SNOW.MODEL. Given the relative success of the $\Delta$SNOW.MODEL in its current version – especially the reduction of the RMSE of $SWE_{\mathrm{pk}}$ by 50-70% compared to ERMs and
*maybe* even down to the $SWE$ error range of thermodynamic models – the probably very costly, but potentially often only very minor improvements when including rain-on-snow should be considered.

"Maybe" is emphasized in the last paragraph because it leads to another discussion and outlook point: An intensive, multi-year model comparison should be performed; at some benchmark sites with fully equipped snow stations and – very important – different methods of $SWE$ measurements, including regular manual observations (with sampling cylinders etc.). Some of
those data sets do already exist, however, a comprehensive comparison of techniques and methods to measure and, in particular, model $SWE$ is lacking. Often the "target variable" is bulk density (not $SWE$), and relative (not absolute) numbers are the only information on observation accuracy one can get, although recently efforts are undertaken (e.g., in the framework of the ESSEM COST Action ES1404; Leppänen et al., 2018a; López-Moreno et al., submitted, ...) – at least concerning measurements. Shouldn't there be more studies, that also comprehensively quantify the abilities of various, especially thermodynamic snow
models to simulate $SWE$? A top-quality comparison between the $\Delta$SNOW.MODEL and thermodynamic snow models is actually difficult to achieve since hardly any numbers for $SWE$ accuracy of thermodynamic models are available. *Maybe* they perform worse than the generously estimated RMSEs of $10$ to $20\,\mathrm{kg\,m^{-2}}$ of this study?

Another discussion point and eventual future development is the refinement of the density parameters $\rho_0$ and $\rho_{\max}$ since, firstly, the $\Delta$SNOW.MODEL reacts quite sensitive on their changes and, secondly, some relations are well known, e.g., $\rho_0$'s





dependence on the climatic aridness or $\rho_{\max}$'s tendentious increase for (very) old snow. Additional calibrations could be performed for very maritime, very dry, polar, or tundra regions as well as for very long-lasting snowpacks. Note, however, all of these adaptions introduce more parameters to the $\Delta$SNOW.MODEL and reduce its generality. Benefits should be evaluated critically. Probably against the overburden load treatment of the $\Delta$SNOW.MODEL, since it is possible that refining the density parameters is more valuable than the special treatment of unsteady compaction due to overburden loads...

Last but not least, looking at current developments in deriving $SWE$ from snow heights, that are monitored with lidar and photogrammetry, the $\Delta$SNOW.MODEL should be considered as – following Smyth et al. (2019) – one of the "potentially [applicable] other snow density models". Lidar and photogrammetry have errors in the order of $10\,\mathrm{cm}$ (Smyth et al., 2019), typically corresponding to $SWE$ errors of $20$ to $40\,\mathrm{kg\,m^{-2}}$. This is in the order or somewhat more than the error of the $\Delta$SNOW.MODEL. Remote sensing derived snow height data are discontinuous through time. The $\Delta$SNOW.MODEL would have to be adapted to

that (which might not be a big issue), though, for the big benefit of being independent from meteorological data and models – and their errors.

## 6   Conclusions

A new method to simulate snow water equivalents ($SWE$s) is presented. It exclusively needs snow heights and their temporal changes as input, which is its major advantage compared to other snow models. It is shown that basic snow physics, smartly im-

plemented in a layer model, suffice to better calculate $SWE$ than snow models relying on empirical regressions. Consequently, the study's null hypothesis (Sect. 1.3) is rejected.

    Gapless snow height records are used to stepwise model the evolution of seasonal snowpacks, focusing on their mass (i.e. $SWE$) and respective load. Snow compaction is assumed to follow Newtonian viscosity, unsteady stress for underlying snow layers by the overburden load of fresh snow is regarded separately, melted mass is distributed from upper to lower

layers, and – eponymous for the model – the measured change in snow height between two observations is used as a precious corrective, though by accounting for measurement uncertainties. The model steps are rather simple, however tricky in details, and all is frankly revealed in this article.

    The $\Delta$SNOW.MODEL mainly bases on Martinec and Rango (1991) and Sturm and Holmgren (1998), and transforms them to a modern R-code, which is available through https://r-forge.r-project.org (*nixmass* package). Other meteorological (aside

$H$) and also geographical input is consequently avoided in the framework of the $\Delta$SNOW.MODEL. Still, calibration of seven parameters is needed. To provide an optimal setting and utmost applicability, data from 14 climatologically different places in the Swiss and Austrian Alps are utilized. This is challenging, since calibration needs multi-year $SWE$ observations as well as consecutive (e.g. daily) snow height readings from the same places. The $\Delta$SNOW.MODEL is calibrated with the help of 71 winters. The validation data set consists of another 73 independent winters. Whereas calibration is rather complex, the

application of the $\Delta$SNOW.MODEL is cheap in terms of computational effort: Deriving a one-year $SWE$ record from 365 snow height values, e.g., only takes a few seconds with today's standard desktop CPUs and can certainly be speeded up significantly.



It is argued that the $\Delta$SNOW.MODEL is situated between sophisticated "thermodynamic snow models", necessitating lots of meteorological and other input, and modest "empirical regression models" (ERMs), relying on statistical relations between $SWE$ and snow height, date, altitude, and region.

– This "interposition" is true in terms of model complexity: The $\Delta$SNOW.MODEL is a semi-empirical multi-layer model but only needs one input variable, which is $H$.

   – Still, the $\Delta$SNOW.MODEL is even less demanding than ERMs: It exclusively needs $H$ records, though gapless ones. No information on date, altitude, or region is required.

   – In terms of universality, this shifts the $\Delta$SNOW.MODEL close to thermodynamic models, and because the $\Delta$SNOW.MODEL
755       simulates individual $SWE$ values – like the important seasonal maximum $SWE_{\mathrm{pk}}$ – as good as $SWE$ averages, it can compete with thermodynamic models at the, e.g., daily level as well (which is not reasonable for ERMs).

   – Not least, the $\Delta$SNOW.MODEL's performance in modeling $SWE$ lies between thermodynamic models and ERMs, albeit close to the very sophisticated ones: Root mean squared errors for $SWE_{\mathrm{pk}}$ (cf. Table 2) are at ca. $23\,\mathrm{kg\,m^{-2}}$ ($\Delta$SNOW.MODEL), at about $60$ to $90\,\mathrm{kg\,m^{-2}}$ (ERMs), and somewhere between $10$ and $40\,\mathrm{kg\,m^{-2}}$ (thermodynamic mod-
760       els).

Given these promising results, the $\Delta$SNOW.MODEL's ancestors Sturm and Holmgren (1998)'s argument, whereby "snow load plays a more limited role in determining the compaction behavior in seasonal snow than grain and bond characteristics and temperature", might be disproved.

The development of the $\Delta$SNOW.MODEL is application-driven. It is therefore not surprising that this study provides no sig-
nificant new findings in snow physics. Still, the $\Delta$SNOW.MODEL seems to be the first model (since long) that takes well known basic snow principles and arranges them in a physically consistent way, while consequently ignoring all potential information except snow height. Not particularly innovative, but remarkably successful. Nevertheless, the synopsis of the $\Delta$SNOW.MODEL and measured data gives significant hints on two important snowpack features (at least for the Alps): Typical mean density for fresh snow ($24\,\mathrm{h}$) seems to be clearly below often assumed $100\,\mathrm{kg\,m^{-3}}$ and a characteristic average maximum bulk density
for seasonal snow (also including ephemeral snowpacks from low-lying places) can rather be found around $400\,\mathrm{kg\,m^{-3}}$ than at often cited $500$ to $600\,\mathrm{kg\,m^{-3}}$, which might be biased by "too alpine" snowpacks for many applications. The $\Delta$SNOW.MODEL is widely usable, but first of all it can attribute snow water equivalents to all longterm and historic snow height records, which are so valuable for climatological studies and extreme value analysis for risk assessment of natural hazards .

*Code availability.* https://r-forge.r-project.org (*nixmass* package)





## Appendix A

The Scaling Module (Sect. 2.2.2) recalculates the viscosity parameter $\eta_0$. This temporary $\eta_0^*(i,t)$ does not only depend on the point in time $t$ (whenever the Scaling Module is activated), but is also different for each layer $i$. The reason is described in the following.

The Scaling Module aims for the condition, that today's model snow height $H(t)$ equals today's observed snow height $H_{\text{obs}}(t)$.

$$H(t) = \sum_{i=1}^{ly(t)} h(i,t) \overset{!}{=} H_{\text{obs}}(t)$$

It follows from Eq. (3) and substituting $x(i,t) = \Delta t \cdot \widehat{\sigma}(i,t) \cdot e^{-k \cdot \rho(i,t)}$:

$$\sum_{i=1}^{ly(t)} h(i,t) = \sum_{i=1}^{ly(t)} \frac{\eta_0^*(t) \cdot h(i,t-1)}{\eta_0^*(t) + x(i,t)} \overset{!}{=} H_{\text{obs}}(t),$$ (A1)

which is a rational function $f$ of the form

$$f(\eta) = \sum_{i=1}^{N} \frac{\eta \cdot h_i}{\eta + x_i}$$

Because $f(\eta)$ has poles at $-x_1,\ldots,-x_N$, the equation $f(\eta) = H_{\text{obs}}$ has multiple solutions. Consequently, this approach – with $\eta_0^*(t)$ being independent from layer $i$ – shows a clear non-physical behavior making it necessary to calculate different $\eta_0^*(i,t)$ for each layer $i$ based on Eq. (A1):

$$\eta_0^*(i,t) = \frac{x(i,t) \cdot h(i,t)}{h(i,t-1) - h(t)}$$

The solution of this issue in the Scaling Module of the $\Delta$SNOW.MODEL bases on the assumption, that observed compaction between $t-1$ and $t$ can be approximated linearly for each layer:

$$\frac{h(i,t)}{h(i,t-1)} \overset{!}{\approx} \frac{H_{\text{obs}}(t)}{H_{\text{obs}}(t-1)}$$

The layer-individual viscosities can be calculated as

$$\eta_0^*(i,t) = \frac{x(i,t) \cdot H_{\text{obs}}(t)}{H_{\text{obs}}(t-1) - H_{\text{obs}}(t)}$$

Substituting those values for $\eta_0^*$ in Eq. (A1) fulfills its precondition, and the modeled equals the observed snow height. The newly calculated $\eta_0^*(i,t)$ are different for each layer – in contrast to the fixed $\eta_0$ defined in Sect. 2.1, which is valid for the whole snowpack (outside the Scaling Module). Note that these new viscosities are only used temporarily in the Scaling Module. They have no analog in reality and can also have negative values, but they are mathematically sound.



*Author contributions.* MW rose and led the project, structured and managed it. He was a key figure in developing and designing the snow model, and he did most of the writing. HS developed, coded and calibrated the model. He wrote the R-package and helped writing the paper, particularly the application example. SG developed early versions of the model and its code.

*Competing interests.* The authors declare no competing interests.

*Acknowledgements.* The authors want to acknowledge T. Hell (Dept. of Mathematics, Univ. of Innsbruck) who substantially helped with the Scaling Module (Sect. 2.2.2 and Appendix A). Thanks also go to the Hydrographic Service of Tyrol (Austria) which provided part of the data. A. Radlherr and J. Staudacher is thanked for vivid discussions and proof-reading.

*Financial support.* This work was embedded in the project "Schneelast.Reform", funded by the Austrian Research Promotion Agency (FFG) and the Austrian Economic Chamber (WKO), in particular by their Association of the Austrian Wood Industries (FV Holzindustrie).

*Review statement.* This paper was edited by NAME and reviewed by NAME and/or NUMBER anonymous referee(s).



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

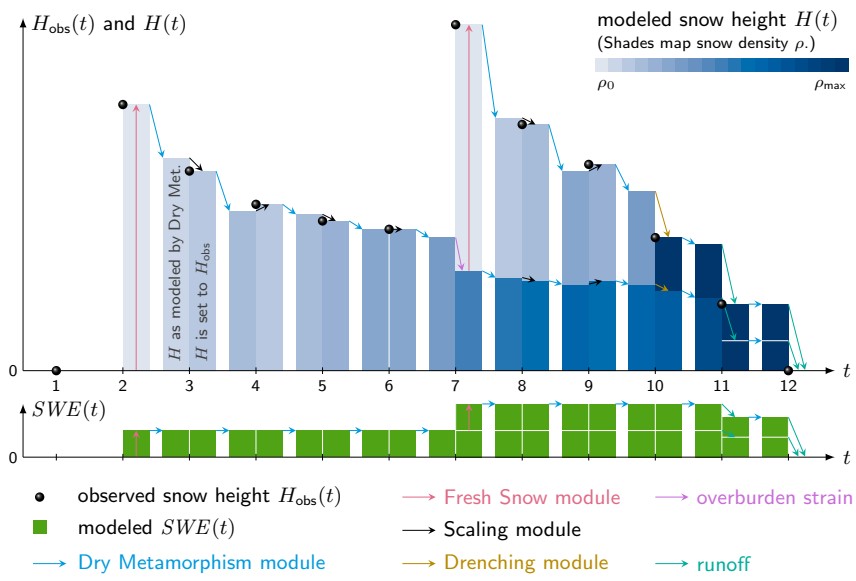

**Figure 1.** Schematic figure of the $\Delta$SNOW.MODEL's principles. See text for more details. At time $t = 1$ no snow is observed (black bullet at $H_{obs}(1) = 0$) and – consequently – no snow is modeled. At $t = 2$ initially no snow is modeled. However, snow is observed ($H_{obs}(2) > 0$) and, thus, model snow height $H(2)$ is set to the observed value by the $\Delta$SNOW.MODEL's *Fresh Snow Module* (pink arrow) and a certain $SWE$ (green boxes) is assigned to this fresh snow layer. For $t = 3$ densification by the *Dry Metamorphism Module* (Dry Met.) is modeled (light blue arrow). Snow density ($\rho$, shown as bluish shades) is enhanced, but $SWE$ stays constant. Still, the model and observation slightly disagree ($|\Delta H| < |\tau|$). The *Scaling Module* solves this issue (black arrow): At $t = 3$ (and also at $t = 4, 5, 6, 8,$ and $9$) modeled $H(t)$ is scaled to equal $H_{obs}(t)$ with respective consequences for the snow density, but not altering $SWE$. $H_{obs}(7)$ is way higher than $H(7)$, the *Fresh Snow Module* builds up a new layer and raises $SWE$ accordingly. The *Fresh Snow Module* also treats the unsteady and strong compaction of the underlying layer(s) due to overburden snow load (purple arrow). At $t = 10$ observed snow height is significantly smaller than $H(10)$, the *Drenching Module* (brown arrows) wettens the layers from top to bottom until $\rho_{max}$ is reached. Figuratively, the layers get water-saturated, however, at $t = 10$ not all layers reach $\rho_{max}$: No mass loss is requested by the model, and $SWE$ stays constant. At $t = 11$ the Drenching Module necessitates mass loss by runoff (cyan arrows) as all layer densities would have to exceed $\rho_{max}$ to fulfill $H(t) = H_{obs}(t)$, but this is not possible in the $\Delta$SNOW.MODEL. All layers are set to $\rho_{max}$ and they "get cut" by an appropriate amount of height and mass, respectively, depending on their thickness: Thick layers contribute more to the mass loss than thin ones. In the end, at $t = 12$, no snow is observed anymore and final runoff is modeled.





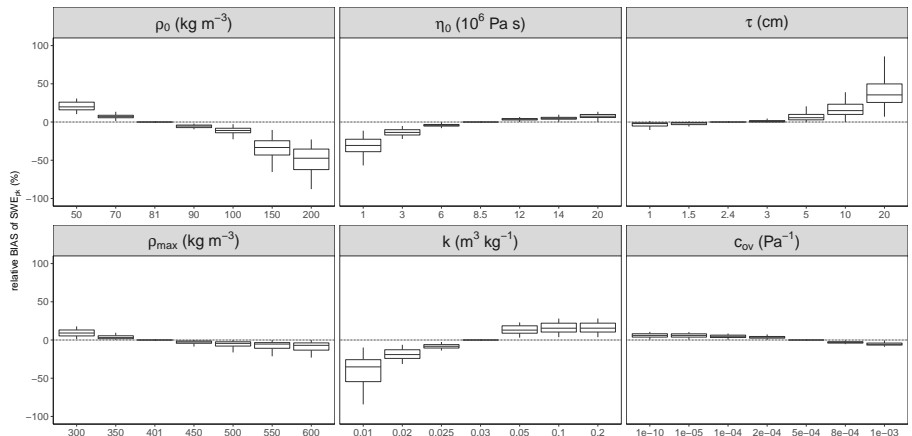

**Figure 2.** Sensitivity of $SWE_{pk}$ to changes in model parameters. The relative bias of $SWE_{pk}$ is defined as the difference between $SWE_{pk}$ with best-fitted values and $SWE_{pk}$ with changed parameters (while all others are kept unchanged), divided by the best-fitted $SWE_{pk}$. The boxes comprise $SWE_{pk}$ of all stations and all years of the validation data set $SWE_{val}$ (73 values) and display medians as well as 25% and 75% percentiles, the whiskers indicate minimum and maximum biases. (Parameter $k_{ov}$ behaves unremarkably – similar to $c_{ov}$ – and is not shown here.) Details and analysis see text.

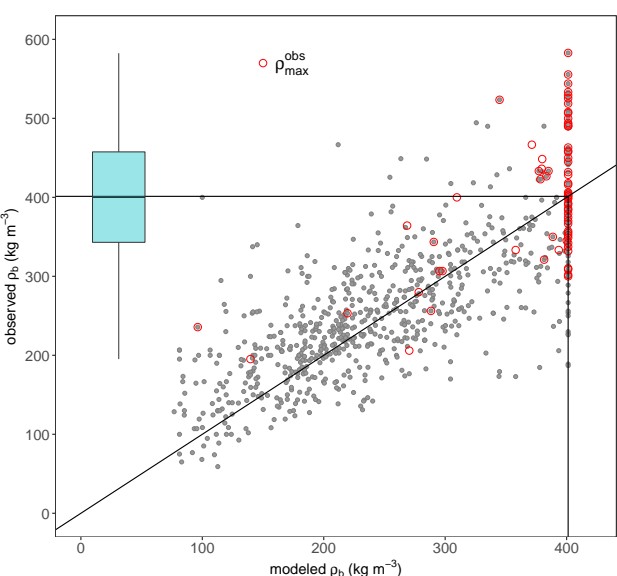

**Figure 3.** Scatter plot of all modeled bulk snow densities $\rho_b$ versus all observed $\rho_b$ from the validation data set. ($SWE_{\text{val}}$, 761 data pairs. Seven observations, which are higher than $600 \, \text{kg m}^{-3}$, were ignored due to implausibility.) Circles reflect the 73 observed yearly maxima ($\rho_{\text{max}}^{\text{obs}}$), most of them occur when also all modeled snow layers are saturated at $\rho_{\text{max}} = 401 \, \text{kg m}^{-3}$. The box plot shows the distribution of $\rho_{\text{max}}^{\text{obs}}$: The median is at $400 \, \text{kg m}^{-3}$. (This round value is somewhat fortuitously and should not be taken too seriously.) The horizontal line compares it to the $\Delta\text{SNOW.MODEL}$'s maximum density at $\rho_{\text{max}}$.



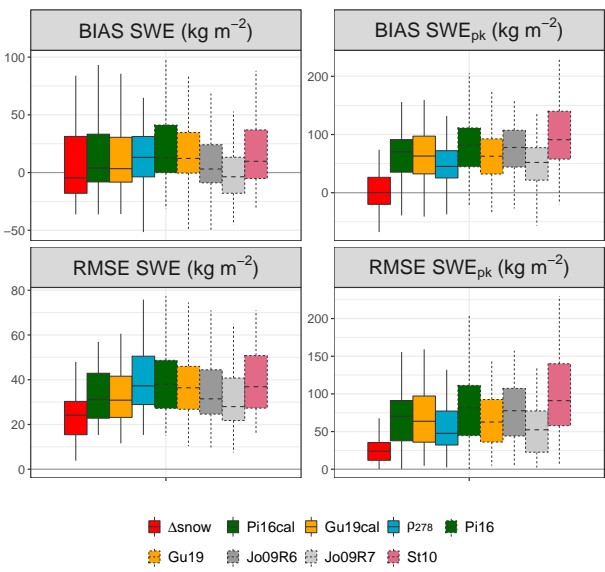

**Figure 4.** Validation results for the model biases and root mean squared errors (RMSE). The plots show the results for the models applied to the $SWE_{\text{val}}$ data. The $\Delta$SNOW.MODEL, Pistocchi (2016), Guyennon et al. (2019), and the "constant density approach" were calibrated with $SWE_{\text{cal}}$ data (Pi16cal, Gu19cal, $\rho_{278}$). Dashed lines indicate the Pistocchi (2016), the Guyennon et al. (2019), the Jonas et al. (2009), and the Sturm et al. (2010) model with their standard parameters (Pi16, Gu19, Jo09R6, Jo09R7, and St10). Jo09R6 and Jo09R7 together illustrate the maximum possible spread of the Jonas et al. (2009) model since Region 6 (R6) and Region 7 (R7) are characterized by the highest and lowest "region-specific offset", respectively. The boxes encompass 761 values (left panels, $SWE$) and 73 values (right panels $SWE_{\text{pk}}$) and spread from the 25%- to the 75%-quantile, the whiskers indicate minima and maxima. The $\Delta$SNOW.MODEL does not behave significantly better for the bias of all $SWE$ (upper left), however, its performance for seasonal maxima $SWE_{\text{pk}}$ (upper right) as well as for the mean errors (lower panels) is very convincing. (Note the different y-axes scalings.)



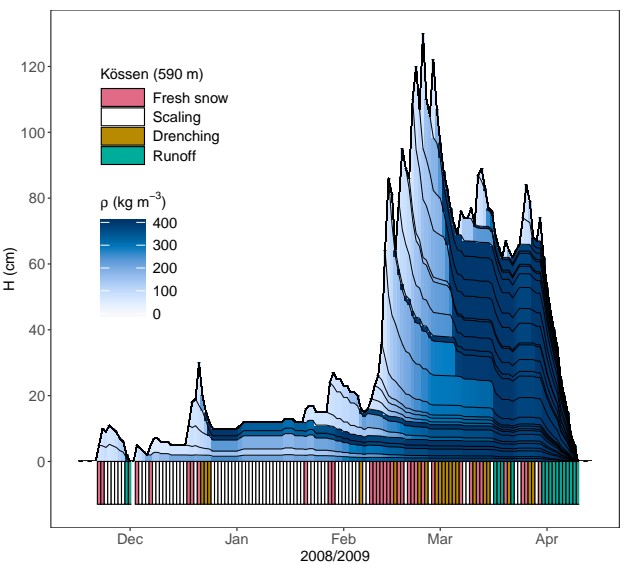

**Figure 5.** The winter of 2008/09 in Kössen, a low-lying but snowy station at the Northern Alps, portrays density evolution (shades) as simulated by the $\Delta$SNOW.MODEL. The gapless, daily snow height record is used as only model input. Three (out of four) modules of the $\Delta$SNOW.MODEL are depicted in colors at the bottom, whenever activated: Drenching, Fresh Snow, and Scaling (The Dry Metamorphism Module is activated at every point in time.) Runoff is a subcategory of the Drenching Module. Note, the $\Delta$SNOW.MODEL is not intended to simulate individual layers. This figure illustrates what happens during the modeling. However, the aim of the $\Delta$SNOW.MODEL is to get daily $SWE$ and $SWE_{\text{pk}}$ right – i.e. *mean* daily bulk density, not layer-individual densities. Descriptions and discussions of some features are given in the text.



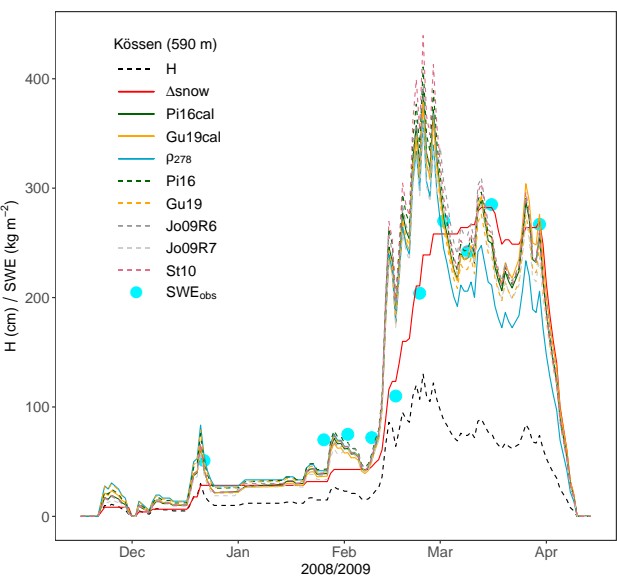

**Figure 6.** $SWE$ simulations and observations (mostly weekly; $SWE_{obs}$) for the winter 2008/09 in Kössen (cf. Fig. 5). Model abbreviations are given in the text and summarized in Fig. 4. See Table 2 for values, and consider the note in its caption. This plot is an illustration of the $\Delta$SNOW.MODEL performance during a distinct winter and outlines important features, which are addressed in the text.



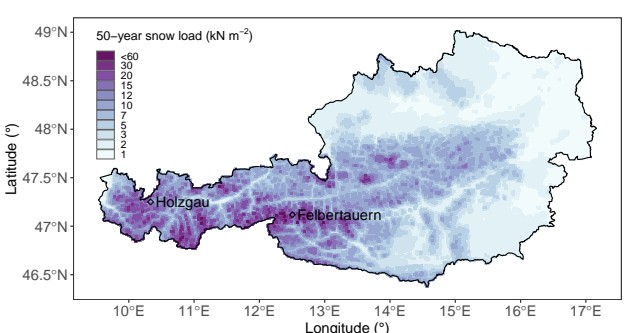

**Figure 7.** 50-year return levels of snow load in Austria. Two stations with $SWE$ observations are outlined for a qualitative validation. This map bases on 214 snow height records, $\Delta$SNOW.MODEL derived $SWE$, and smooth spatial modeling of their extremes. See text for details.

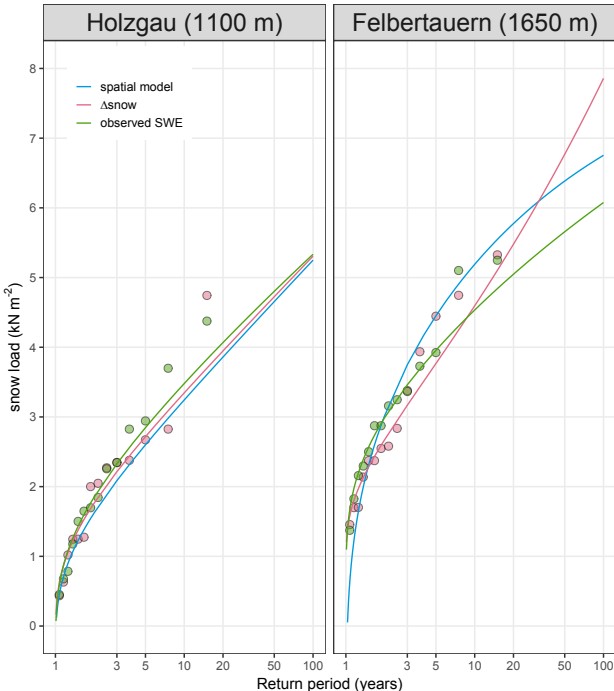

**Figure 8.** Return levels of snow load at stations Holzgau (left) and Felbertauern. Return periods in years are shown on the logarithmic x-axis. The blue line shows return levels obtained with the spatial extreme value model, pink bullets and lines depict yearly maxima and the GEV fit of $SWE$ values modeled from daily snow heights with the $\Delta$SNOW.MODEL, and green colors represent yearly $SWE$ maxima and the corresponding GEV fit from (ca. weekly) observations.





**Table 1.** The seven parameters of the $\Delta$SNOW.MODEL. The last column depicts model sensitivity to changes in the density parameters. The respective gradients are means over the whole calibration ranges. Detailed information is found in the text.

| Parameter $par$ | unit | optimal value | calibration range | literature range | sensitivity $\frac{\delta SWE_{pk}\,[\mathrm{kg\,m^{-2}}]}{\delta par}$ |
|---|---|---|---|---|---|
| $\rho_0$ | $\mathrm{kg\,m^{-3}}$ | **81** | 50-200 | 75[a], 10-350 (70-110)[b] | +0.37 (+0.50[†]) |
| $\rho_{\max}$ | $\mathrm{kg\,m^{-3}}$ | **401** | 300-600 | 217-598[c], 400-800[d] | +0.24 |
| $\eta_0$ | $10^6\,\mathrm{Pa\,s}$ | **8.52** | 1-20 | 8.5[a], 6[e], 7.62237[f] | not calc. |
| $k$ | $\mathrm{m^3\,kg^{-1}}$ | **0.0299** | 0.01-0.2 | 0.011-0.08[a], 0.185[g], 0.023[e,f], 0.021[h] | not calc. |
| $\tau$ | cm | **2.36** | 1-20 | - | not calc. |
| $c_{ov}$ | $10^{-4}\,\mathrm{Pa^{-1}}$ | **5.10** | 0-10 | - | not calc. |
| $k_{ov}$ | - | **0.379** | 0.01-10 | - | not calc. |

[a]Sturm and Holmgren (1998), [b]Helfricht et al. (2018) with range for means in brackets, [c]Sturm et al. (2010), [d]Paterson (1998), [e]Jordan et al. (2010), [f]Vionnet et al. (2012), [g]Keeler (1969), [h]Jordan (1991). See Sect. 2.3 for more details. [†]The value in brackets is the gradient taken from the smaller window between 70 and 90 kg m$^{-3}$ (cf. Sect. 3.1.1).





**Table 2.** Overview on $SWE$ accuracies of different methods and studies. The numbers of this study are the median values, which are also depicted in Fig. 4. The numbers in brackets represent the results for the example portrayed in Figs. 5 and 6 from station Kössen in 2008/09. Note, the performance of the ΔSNOW.MODEL of the example is quite ordinary, while other models do better on average. Units are $\mathrm{kg\,m^{-2}}$ except last column, which is in days.

| source | model (version) | $SWE$ BIAS | $SWE$ RMSE | $SWE$ MAE | $SWE_{\mathrm{pk}}$ BIAS | $SWE_{\mathrm{pk}}$ RMSE | $SWE_{\mathrm{pk}}$ BIAS [d] |
|---|---|---|---|---|---|---|---|
| this study | ΔSNOW.MODEL | −4.0 | 23.9 (21) | 19.5 | 2.3 (−3) | 23.1 | 0 |
| | Gu19cal | 4.0 | 31.3 (43) | 24.4 | 67.3 (93) | 70.8 | −6 |
| | Pi16cal | 5.3 | 32.9 (47) | 25.6 | 71.0 (106) | 72.2 | −6 |
| | Jo09R7 | −2.0 | 26.6 (41) | 21.9 | 56.5 (74) | 58.1 | −2 |
| | St10 | 17.6 | 37.1 (57) | 28.5 | 95.1 (154) | 95.7 | −11 |
| | $\rho_{278}$ | 14.8 | 38.1 (51) | 31.2 | 47.7 (77) | 53.5 | −16 |
| Guyennon et al. (2019) | Gu19 | | | 49.2 | | | |
| | Pi16cal | | | 50.6 | | | |
| | Jo09cal | | | 48.5 | | | |
| | St10cal | | | 51.0 | | | |
| Jonas et al. (2009) | Jo09 | | | 50.9 − 53.2 | | | |
| Sturm et al. (2010) | St10 ("alpine") | 29 ± 57 | | | | | |
| Vionnet et al. (2012) | Crocus | −17.3 | 39.7 | | | | |
| Langlois et al. (2009) | Crocus | −7.9 to − 5.4 | 10.8 − 12.5 | | | | |
| | SNTHERM | 9 to 18.1 | 18.3 − 19.3 | | | | |
| | SNOWPACK | −0.1 to 5.6 | 7.4 − 14.5 | | | | |
| Sandells et al. (2012) | SNOBAL | | 30 − 49 | | | 17 − 44[a] | |

[a]This is not RMSE of $SWE_{\mathrm{pk}}$, but RMSE "from establishment of snowpack to $SWE_{\mathrm{pk}}$".