# Peer review of "Snow Water Equivalents exclusively from Snow Depths and their temporal Changes: The $\Delta$ SNOW.MODEL"

_Hydrology and Earth System Sciences, 2020_

## Referee Comment (RC1) · Anonymous Referee #1 · 28 Apr 2020

Dear authors,

congratulations to your impressive paper manuscript. It represents a huge piece of hard work. You have invested significant efforts to publish your ideas and the comprehensive dataset you can utilize. I liked reading your paper and following your argumentation. However, prior to publishing it as final revised paper I recommend a general revision and sharpening of the focus of the manuscript, as well as an improvement of the English.

General comments:

My main point (1) is that the entire paper reads too much like a multi-faceted story

around various aspects of model development, calibration/evaluation, model intercomparison and historical model review. For the final version, the manuscript would benefit from a general re-shaping to a concise description of the particular innovation you developed. The manuscript would also profit from being significantly shortened.

The second issue (2) is the language. I am not a native speaker myself, but I can tell that many phrases and terms are untypical for the particular language that is required for a scientific paper. Maybe you can find a native speaker who can carefully check the final version of the text to make it a clear scientific argumentation with proper formulations.

Finally (3), I added a list of general comments related to particular aspects and formulations in the text.

Despite all criticism the model approach you develop is well worth to invest the required energy into the manuscript to make it a final revised paper.

Specific comments (ad 1):

(i) I recommend to decide what the main focus of your paper is, and delete anything else that is not needed. Why? After reading the paper, it is not entirely clear to the reader if the main focus is (i) the introduction of a new model, (ii) the improvement of an existing model, (iii) a parameterization exercise, (iv) a model intercomparison, or (v) about snow loads. The main focus of the paper changes. I would recommend to limit the aim to one, and to reduce the description of objectives to what leads to the one aim. My personal suggestion would be: the significant innovation is the fusion and improvement of the Gruber/Sturm-Holmgren/Rango-Martinec approaches (as you state in lines 414-416): You could build the paper upon the differences between these approaches and the respective innovation in your model (-> exponential function instead of power law for compaction etc.), and then evaluate the new model version against (i) the other three approaches, and against (ii) observations.

(ii) The paper manuscript lacks a clear description of the data that is used for calibration and for application of the new model version, including a map of the station locations.

(iii) The role of rain-on-snow is crucial and deserves more attention, and not just an announcement for further model improvements. See also (vi)

(iv) You better strictly separate physical process descriptions from code structure descriptions. Skip the latter if your paper is not intended to be a model code introduction (or move it to the appendix). I strongly recommend to leave any code structure elements out of the main body of the manuscript. Better set the focus on the general rules that your model follows in representing the snow layers and their changes.

(v) Your null hypothesis, and its rejection, is not required. You can omit it.

(vi) Your new approach could easily be combined with daily temperature and humidtiy recordings; these are available in many regions of the world. Wouldn't it be most interesting to use these as auxiliary data to derive precipitation phase and melting conditions (and hence, solve some important issues that you address)?

To say it again: I strongly want to motivate and encourage you to elaborate on your manuscript and submit an improved version. The material you already have and the newly developed model version will be a significant and valuable contribution for the snow modelling community.

Ad (2):

Expressions like the following examples should either be replaced with clear scientific statements, or omitted.

- 212: "figuratively spoken the Dry Metamorphism Module acts "over night""

- 284-285: "equations 3) and 4) . . . make no claims of being particularly precise"

- 576: "(Still, lots of them!)"

- 736/737: "The model steps are rather simple, however tricky in details, and all is frankly revealed in this article."

Technical corrections (ad 3):

I started to comment on how to improve the text with the abstract. Then I stopped, because I saw that the manuscript would benefit from substantial language improvement, and this should be better done by a native speaker. Nevertheless, here is what I can come up with for the abstract:

- 1: in Austria?

- SWE = synonymous for snow water equivalent

- 4: skip "fields like"

- 5: better "the respective" than "those"

- 7: better ". . . needs a continuous time series of snow heights without gaps as input"

- 8: better ". . .with arbitrary temporal resolution"

- 12: better make two sentences here; the first is about a general model issue, the second belongs to the particular model application in your paper

- 13: "winters" do not "act". Better something like ". . . data from 73 winters is used for validation"

- 14: replace "squared" with "square" (several locations in the manuscript)

- 16: better ". . . and even thermodynamic . . . do not necessarily perform better".

- 18: Delete "Not least", replace "on . . ." with ". . . of SWE measurements to modeling"

The following comments are therefor mostly suggestions of further improvement of the content of the paper:

- 29-30: "remote sensing" to measure H: which techniques/sensors?

[Figure]

- 25-48: main point missing, should be clearly fomulated: SWE recordings require consideration of the layering of the snow (density can be very different in each single layer), whereas height is a bulk measure and simple to take

- 54: "hydrological", "agricultural", yes, but many more...!

- 70: mainly precipitation, temperature, humidity, wind speed, radiative fluxes and, for some models, air pressure

- 71: what means "... many longterm H series ... do not come along with..." in this context?

- 68-77: better do not put longer parts of text in brackets. Either what you say is important, then it belongs to the manuscript. Or not: then delete it

- 64-77: here I miss the series of SnowMIPs, and the recent papers about snow model performance

- 107: correct "altemative"

- 119: is "snow depth" the same as "snow height"? If you use both in the text, then explain the difference

- 125-135: hence, the presented new model "ΔSNOW.MODEL" is an update of the Gruber-model, based on the Rango/Martinec - approach (exponential function instead of power law for compaction)? See my general comment above

- 141-142: can't find "nixmass" at https://r-forge.r-project.org. All links should be checked if they work

- 149: "section 5 and 6 discuss ...": no, the sections themselves do not do anything. Maybe better: "... in section x and y ... is discussed"

- 157-160: don't switch between physical processes and code structure, same for line 167. See my main comment (1) above: if the paper is not about the new code (and I

suggest to not make a model development paper out of it), then you can omit the model structure (i.e., all that has to do with the modules)

- 177: a "Module" cannot combine anything. Maybe "in . . . module, xxx and yyy are combined"?

- 196-197: again, separate model description from code structure ("Dry Metamorphism Module")

- 200: don't repeat (already stated in lines 89/90 and comes again in 762)

- 228: what happens in the model if an intensive rain-on-snow event occurs? This seems to be of importance and comes too short in the manuscript. See main comments above

- 236: in case $\Delta H$ (t) $> +\tau$, a rain-on-snow event cannot have occured, or what did I oversee here?

- 242-243: again, don't mix processes with code structure

- 355: probably better write "viscosity at which equals zero"

- 380: were the observations all by the Hydrographic Service of Tyrol, also the ones in the Southern Alps?

- 512: again, better "root mean square error"

- 526: are there other calibration/validation data available in the literature (for the models by Jonas et al. (2009) and Sturm et al. (2010))?

- 545: again, better "root mean square error"

- 558: calibrating "the models" by Pistocchi (2016) and . . .

- 560: ". . . Sturm et al. (2010)'s method probably suffers from the handicap of being calibrated with data from the Rocky Mountains": the model does not suffer from anything. It just was calibrated elsewhere; avoid attributing such values

- 569: "well beaten by . . .": avoid such formulations generally

- 685: what is a "Fréchet-like distribution"?

- 695: use temperature/humidity, almost everywhere available and easy to interpolate? See my general comment above

- 709/710: "Shouldn't there be more studies, that also comprehensively quantify the abilities of various, especially thermodynamic snow models to simulate SWE¿' The discussion/outlook section is not the place to ask such questions. And, moreover, there are the SnowMIPs and recent papers about snow model intercomparisons/performances

- 761-763: "Given these promising results, the ΔSNOW.MODEL's ancestors Sturm and Holmgren (1998)'s argument, whereby "snow load plays a more limited role in determining the compaction behavior in seasonal snow than grain and bond characteristics and temperature", might be disproved."?! I wouldn't say this, because snow load is one of the origins that change grain and bond characteristics, as is temperature, humidity gradient etc.. The compaction behavior finally depends on the magnitude and rate of change of each of these forces, and their interplay. Better skip this entire line of argumentation here, there is no need for it neither any gain.

Figure 1: Better do not assign the same type of symbol (arrows) to two very different things at the same time: program modules (= code structure) and hydrological processes (like runoff). The codes structure (name of the modules) can be deleted here (and in the caption which is way too long). Your new model version is mostly rule based, hence how about a decision tree to illustrate its functioning?
* * *

---

## Short Comment (SC1) · 9 May 2020

Congratulations on this great piece of work, which can have many applications.

I tried out the R-package and it works smooth for the HS series I have. However, you should state somewhere, that the code should be run separately per hydrological year (I initially tried it with a 30 year period, and it was very very slow).

Regarding your manuscript, I miss some structure. Methods, results, and discussion are all mixed together. It might be worth to split at least methods from results&discussion, and the methods into your describing the delta snow, and the rest

that you have done.

Regarding your methods, I have to advise against your approach of splitting into calibration and validation by odd/even years. Such a procedure should always be fully randomized. Given your rather limited amount of data, I strongly suggest to apply a cross-validation approach instead of pre-defining two subsets. Also an 80-20 split might be the better option to have more data available for training (and then run it 5 times, to have each observation used for both training and testing, reducing selection bias). And then, if you want, but I definitely can recommend it, you could repeat this whole procedure (randomly splitting data, training, validating) 100 (or better 1000) times, in order to get an estimate of the uncertainty introduced by your data set. This concerns especially the uncertainty on your calibration parameters and validation estimates. Another benefit would be that estimates are more robust (when it come to generalizing) with such a fully randomized approach.

Unfortunately, you did not have the chance to validate your model against the thermodynamical ones. Would love to see an intercomparison of the existing models, both ERM and thermodynamical. Maybe for another study...

Otherwise, well done and looking forward to applying your model!

---

## Referee Comment (RC2) · Anonymous Referee #2 · 12 May 2020

This is a remarkable piece of work. Congratulation! I liked reading the paper, which has a good structure and an easy to read language. The study is definitely worth to be published in HESS, after the following major points have been addressed:

1) One uncertainty in comparing measured SWE with parametrized SWE from a nearby snow depth measurement stems from the fact that the sum of the heights of the measured SWE samples does usually not correspond to the daily measured snow depth at a graduated pole. This can be due to e.g. to uneven ground or uneven snow distribution (see also chapter 2.4.1.2 in https://www.wmo.int/pages/prog/www/IMOP/publications/CIMO-

Guide/Prelim_2018_ed/8_cryo_2_en_MR.pdf ). Moreover, since the SWE measurement is a destructive method the exact location throughout the winter is changing. For all these reasons the SWE measurements throughout the season and years should be referenced to fixed graduated pole of the daily snow depth measurement by deriving SWE from multiplication of the measured bulk density with the daily measured snow depth at the graduated pole. Please describe how you handled this problem?

2) How can you provide uncertainty statistics (in mass and timing) about SWEpk when your manually measured reference SWE is only measured weekly or bi-weekly. The manually measured SWEpk may have missed the real SWEpk?

3) Please also provide relative error measures. Only relative errors allow to compare results for shallow or deep snow packs.

4) The statement "hardly any numbers for SWE accuracy of thermodynamic models are available" demonstrates that the authors should have done a more thorough literature research. Therefore, please also include at least parts of the results of the following papers:

https://arc.lib.montana.edu/snow-science/objects/issw-2002-353-360.pdf

https://agupubs.onlinelibrary.wiley.com/doi/full/10.1029/2008JD011063

https://www.the-cryosphere.net/9/2271/2015/tc-9-2271-2015.pdf

Specific comments:

20: I suggest to use the official abbreviation HS throughout the paper. See: Page 10 in https://unesdoc.unesco.org/ark:/48223/pf0000186462

21: why "areal" density. Use bulk density or density!

69: I suggest to add the results of the following two studies to table 2:

- doi:10.5194/tc-9-2271-2015

- Rasmus, S., Gronholm, T., Lehning, M., Validation of the SNOWPACK model in five different snow zones in Finland, 2007, Boreal Env. Res., 12(4), 467-488.

120: Either "depth" or "height", I would suggest depth throughout the paper.

124: The approach of Martinec and Rango (1991) was introduced for SWEpk. Rohrer and Braun (1994) extended this approach for daily SWE (https://doi.org/10.2166/nh.1994.0020 ).

159: I suggest to change "fresh snow" to "new snow" throughout the paper according to Fierz et al. (2009).

306: "model snowpack"?

309: Where does the 450 kg/m3 come from? Rohrer and Braun (1991) already used 450 kg/m3.

355: "the viscosity at ïĄš equals zero"?

364: The last two parameters, cov and kov, determine...

503: See also general comment 1

506: "with filling height having the largest influence"? Please elaborate!

576: Please elaborate "diverse measurements"?

581: Omit "now"

582: Unfortunately, there are not the same colors used!

614: "does not play a role" = could not be detected?

664: It should be mentioned in the figure caption, that the gridded information is based on the model Snowgrid. Moreover, it is not all clear if the SWE values used for the spatial extreme model are directly from Snowgrid model output or converted form Snowgrid snow depth by the DeltaSnow.Model?

667: The average reader has no idea that 25 kN/m2 is much too high!

691: You may also mention that your approach only allows either snow fall or melt, but not both, although in reality this happens often. Moreover, what a about mass loss by sublimation?

708: I suggest to replace Leppänen et al. (2008b) with https://doi.org/10.1002/hyp.13785 throughout the paper.

732: Gapless snow depth records are required. . .

768: It is not right to make a general statement like "Typical mean density for fresh snow(24 h)seems to be clearly below often assumed100 kg m$-3$" as long as you can only consider snowfall, when SWE increases, i.e. you miss snowfall events, when concurrent melt occurs, whose mass loss is larger than the mass gain by snowfall (see also comment on line 691). Moreover, you miss all small snowfall events, which are smaller than your uncertainty measure of 2.36 cm.

---

## Author Comment (AC1) · 24 May 2020

We want to thank Anonymous Referee #1 (AR1) for her/his supporting and motivating words as well as for the valuable and positive criticism. The following is our general answer, detailed changes in the manuscript following AR1's suggestions will be outlined in a separate comment together with the revised paper.

First we want to address the referee's "main point (1)", according to which the paper was written like a story, it should be re-sharpened and shortened to a concise description of the particular innovation:

[Figure]

We can understand this criticism, but the narrative style was chosen consciously. Assumingly, the reason why it was "liked to read" (cit. AR1 and AR2) and written in an "easy to read language" (cit. AR2) grounds on this way of writing. We will define the aim of the paper more precisely in the revised manuscript's motivation.

Ad "specific comment (ad 1)", (i):
We present a new model, a new method to simulate SWE. We will try to emphasis this even more during the revision, although this is already literally stated in the abstract, in the header of section 1.3 and (twice) in its body as well as at the very beginning of the conclusion section. Maybe the frank acknowledgements (at two occasions in the text) of not having improved physical knowledge on snow, leads to the impression of not having developed a new, standalone modeling approach. The model of course originates from former developments, but it cannot be seen as "the improvement of an existing model" (cit. AR1). Therefore, we cannot linearly build the paper from "Gruber/Sturm-Holmgren/Rango-Martinec approaches" – as AR1 suggests to do. Still, we will sharpen the manuscript in this respect: (1) "model structure and audacity to consequently use only snow height in digital times" is taken from Gruber2014 (who takes the power law from Martinec/Rango1991) and (2) "exponential compaction" is taken from Sturm/Holmgren1998. Unfortunately, it is not trivial to compare the $\Delta$SNOW.MODEL with Martinec/Rango1991 and Sturm/Holmgren1998. The same is true for Gruber2014, whose code suffers from some major bugs. Martinec/Rango1991 as well as Sturm/Holmgren1998 do not provide a method how to deal with declining snow heights, the question how mass loss is computed is not answered in their papers. This strengthens our argumentation to provide a thorough explanation how our new model is coded (see below). We do compare the $\Delta$SNOW.MODEL with observations, see whole Sect. 3. Involving "a parameterization exercise" and "a model intercomparison" (cit. AR1) is an absolutely necessary part of every model presentation. Omitting these parts would understandably lead to loud criticism. The reason, why parametrization and comparison are found in the results(!) section, is given in the motivation section (1.3). We don't understand, why the paper's focus should be on "(v) snow load" (cit.

AR1). Section 4 is titled "Example of Application", which makes clear that it is not focal, but completes the circle to proposed usability (Sect. 1.3).

Ad "specific comment (ad 1)", (ii):
The description of the data is provided in the cited sources (Gruber2014 and Marty2017). AR1 also criticizes the length of the manuscript. We agree on being lengthy at times and try to shorten the revised version. Including data description, although this information can be found elsewhere, is perceived as contradictory. Nevertheless, we will provide a map in the revised version's appendix.

Ad "specific comment (ad 1)", (iii):
See (vi).

Ad "specific comment (ad 1)", (iv):
The authors do not really understand this criticism. We describe the physics of our model as well as its implementation in the code. That is how modern, open modelling approaches should be introduced. This is not a paper on new physical insights, but it describes how well-known concepts are used, rearranged and set together ("coded") in order to get a new model.

Ad "specific comment (ad 1)", (v):
AR1 is right, this is overweening since we do not have a clear physical, experimental problem to be solved. We omit it.

Ad "specific comment (ad 1)", (vi):
We agree, the new approach could easily be combined with temperature and humidity. As the title already defines ("exclusively"), the $\Delta$SNOW.MODEL should be able to simulate SWE only (!) from snow height. The motivation for the consequent dispensation on further input variables is broadly given in Sect. 1. Including further variables could be future work but – admittedly – not a very innovative one, because such snow models do already exist. During the development of the new approach it was also tried to filter information about rain-on-snow events solely from the snow height records. As others

before, we failed to find a convincing way to do it. We could have changed our plans and include further input variables (which often do not exist, see Sect. 1) to parametrize rain-on-snow, but we decided to ignore it and get surprisingly good results although not explicitly incorporating rain-on-snow. This issue is discussed in Sect. 2.2.3 and 5. The authors will try to formulate all this more precisely for the revised version.

Ad (2)
In the opinion of the authors, these expressions foster readability (which was appreciated by AR1 him-/herself and AR2). We agree, they are not "clear scientific" and will replace them.

Ad (3)
The authors are no native speakers. Nevertheless, we are convinced that the text is written in a proper way: Firstly, there is the somewhat contradictory statement of AR1 ("liked to read"), and AR2's likes the "easy to read language". Secondly, there is hardly any misunderstanding to be eliminated based on AR1's review and not a single typing error was found. Thirdly, a great portion of the requested "technical corrections (ad 3)" are language corrections that could be debated about. We happily observe that AR1's criticism does not involve any objective features of the methods (i.e. the model) and the results, but only the structure and the language of the paper. The latter is quite subjective and the authors argue that this does not justify a major revision.

in detail:
-1: around the globe
-3: SWE is not "synonymous" for snow water equivalent, but SWE is "an abbreviation" of snow water equivalent. However, SWE can be used as a synonym for snow load and snow mass (at least in this paper). That's what is written on line 3 in a grammatically correct way. In the revised manuscript we put SWE in brackets to make it more clearly.
-4 and 5: changed accordingly
-7 and 8: Our intention was to make short and clear sentences in the abstract. We will formulate this differently for the revised version.
-12: We agree, this is a multi-clause sentences, which should be avoided here. Changed accordingly.

-13: changed accordingly (although probably readability suffers)

-14: Both is correct. We discussed it during the writing process and decided to use "squared". Now we change to "square".

-16: changed accordingly

-18: We skip the whole sentence.

-29-30: See five references at the end of the sentence.

-25-48: This is only true for small sampling cylinders (where a pit has to be dug), but not for big snow tubes (see reference). We will add a respective statement.

-54: changed to "hydrological, agricultural, and many other applications"

-70: changed to ", mainly precipitation, temperature, humidity, wind speed and radiative fluxes."

-71: Often there is only a H record, no associated T, rH etc. record. Changed for better understanding.

-68-77: We omit the brackets.

64-77: The requested references will be embedded in the revised manuscript.

-107: There is nothing to correct. Maybe a ligature issue?

-119: "Snow depth" is only used in word-for-word citations when the original author(s) used it as well. At all other occasions we used "snow height" in the manuscript.

We argue "snow height" and "snow depth" are the same and widely used interchangeably, but "snow depth" seems to be the more accepted term. The decision to use "snow height" was not made thoughtlessly: The quantity is measured from ground upward to snow surface and – in this respect – it is a matter of "height" (not "depth", which would be measured from snow surface downward to the ground). Moreover, it is commonly symbolized with "HS" (not least following Fierz2009) and with "H" in this paper. This abbreviations better fit to "snow height".

As AR2 clearly suggests to use "snow depth" (instead of "snow height") and "HS" (instead of "H") we will change the naming for the revised manuscript in the respective

way. This will also change the title.

-125-135: We agree to AR1, this could be misunderstood. We will find other words here. We are convinced that the $\Delta$SNOW.MODEL can be regarded as a stand-alone model.

-141-142: Sorry for that. In the meantime "nixmass" was also ported to CRAN and can easily be found there.

-149: changed accordingly

-157-160: See above (Ad "specific comment (ad 1)", (iv)). The authors cannot follow this criticism and related suggestions. It is the core purpose of the paper to describe how physics is converted to model code. Therefore, we have to describe how specific physical processes, the treatment of measurement and model uncertainties etc. are implemented in the modules of the $\Delta$SNOW.MODEL.

-177: changed accordingly

-196-197: Kept unchanged, see above.

-200: We leave this here, but omit it at the other two locations.

-228: As stated above, the rain-on-snow issue will be sharpened during the revision.

-236: We agree, this is misleading. We will delete the words in brackets.

-242-243: At this location it gets especially obvious how thoroughly we try to describe in which way the physical concepts (taken from Jordan (2010) here) are converted to model code. This is what we think is the core of a transparent introduction of a new model, we therefore do not change the text here.

-355: As stated with quotes and respective reference this definition of $\eta_0$ was literally taken from Sturm and Holmgren (1998). It is changed to "viscosity at [which] $\rho$ equals zero".

-380: Yes, because the Hydrographic Service of Tyrol is also responsible for the region East Tyrol. Its southernmost mountains belong to the "Lienzer Dolomiten" and the "Karnische Alpen", which indeed are parts of the Southern Alps. There will be a map in the revised manuscript's appendix which will clarify.

-512: Changed accordingly.

-526: The authors of the manuscript are not aware of any other available calibration/validation data for those two models.

-545 558 560 569: Changed accordingly.

-685: Not essential. We will omit "due to the Fréchet-like distribution"

-695: Unfortunately not. For most of the longterm and historic snow height records we do not have temperature, let alone humidity etc. (see Sect. 1) – except reanalyses with respective uncertainties. We will sharpen the text for the revised version. See also arguments above.

-709/710: We will omit the question. However, concerning SWE(!)-related studies we still think there is room for more.

-761-763: Changed accordingly.

-Figure 1: The authors think that is the key message of the figure. It shows how physics and model code are combined, it will be tried to graphically separate the two (code/hydrologic processes) during the revision.

We tried to do it as a block diagram but this would take too much space. Moreover, a slender block diagram also needs a longer than average caption. So we stayed with the bar chart and the – admittedly, but consciously – quite extensive caption. Figure 1 (plus its caption) can stand on its own and describes the $\Delta$SNOW.MODEL to a great extent.

---

## Referee Comment (RC3) · Anonymous Referee #3 · 25 May 2020

Snow water equivalent (SWE) has dominating significance for snow hydrology and climatology, however, it is difficult to measure and most available records are based on snow height (H) which is simpler to retrieve. This manuscript (MS) presents an interesting approach to estimating SWE from snow height records and addresses hence an important task. The method described here is not entirely new and the authors frankly state the source of their inspiration, but repeatedly sell their implementation as a new method. There is enough innovation to warrant a publication. and in my view, the MS is served better without this persuasive layer of arguing for newness.

Nevertheless, considerable revisions would be needed to shorten the somewhat

lengthy MS and to clarify its presentation. Currently it is not focused in its objectives and it is hard to understand at a conceptual level: what does the paper want?

a) Describe new method and proof the concept?

Or

b) Show its superiority wrt other existing methods?

This is somewhat unclear but the MS covers aspects of both a and b, without satisfying each. The authors should make a decision and revise the MS following a clearly laid out motivation.

SWE and H are related by the bulk snow density rho_b, such that SWE = rho_b * H

Changes are either by adding/ subtracting mass from the test volume, i.e. by snowfall, deposition of wind blown snow, meltwater runoff or wind erosion or due to gravitational compaction or a combination.

The MS would gain if authors first picture their understanding of the process in detail. This would help to clarify descriptions of their own methods and those of others. If the processes were described in advance it would be easier to put a name on things and state which assumptions are made in different methods.

The presented method is based on deriving SWE from H, and falls therefore back to estimating rho_b. In this MS, assuming an initial new snow density, the evolution of rho_b is considered due to compaction (distinguishing two regimes for dry and wet conditions). Deviations between simulated and observed H are assigned to represent mass changes (snowfall, meltwater runoff).

The authors repeatedly make a point out of that their method only uses H as input and appear allergic to including auxiliary meteorological data which could be useful to improve the model. Their argument of data sparsity in high mountain regions is valid, but one may wonder whether snow height records actually are more widespread

available than for instance temperature measurements?

I recommend re-structuring the content of the MS to clearly distinguish between objectives, description of methods, results and discussion. Objectives and conclusions need to be separated: the presentation of a null-hypothesis (L132) followed by the conclusion that it can be rejected, before even the method is introduced, appears a persuasive presumption that is inappropriate in a scientific paper. Instead, the authors should describe what would be needed to reject the null-hypothesis. And if this is the guiding objective in this study, it should be stated "the null-hypothesis" not "a possible null-hypothesis", be clear about your aims!

I found the description 2.1 a wordy mixture of discussion and the actual description. This should be better separated. In this section please just describe how your model works, and discuss choices and their consequences afterwards in the discussion section.

The "results" section (sec 3) describes the outcomes of a parameter sensitivity analysis and the results of the model application are described in a separate section 4, the discussion section then is kept very short. The presentation appears somewhat unclear and merges discussion with presentation of results (especially sec 3).

I recommend to clearly separate discussion material from the presentation of results and moving them to the expected places: lot of sec 3 is discussion material and should be moved there. The discussion should then comprise several parts: parameter uncertainty and uncertainty due to model structure (the latter part is basically missing).

As a matter of fairness, it should be stated that the physically-based "thermodynamic" snow models do describe dry snow compaction according to Newtonian viscosity, though with a transient viscosity. Currently, the MS leaves the impression the usage of viscosity is a unique, ingenious feature of this model. The crucial statement "deltaSnow.Model "rearranges" existing components in a physically consistent way" is a bit hidden and should be more prominent.

The model description would gain from a short description of the procedure and possibly a flow-chart illustration to help the reader understand the procedure. Currently this information is a bit lost in the details.

Specific comments: Throughout the MS: in written language, spell out to avoid the contractions "it's", "don't", "shouldn't", etc

Language should be direct and in complete sentences. Please remove the frequent side remarks in parentheses, Examples: L37, L66, L99, …In some instances, the side remarks are marked by hyphens and sometimes by brackets. L73ff; why is this in parenthesis? if you have to say something state it.

L8: the requirements of a gapless record together with the statement that temporal resolution does not matter does not make sense without specifying "gap" in contrast to "temporal resolution". Records are not continuous but come at discrete time intervals, which could be seen as gaps.

L29; "sonic or laser distance ranging"

L44ff: no need to introduce acronyms such as GPR, CRNS, or GNSS since they are not used

L49: this is not true. Passive microwave radiometry has been used since long to monitor SWE

L75ff: this is an odd conception to consider statistical models as "thermodynamic"…seems like everything else than deltaSnow is considered a thermodynamic model?

L158 "deformation strain" is an odd expression. Strain is a description of deformation.

L168ff the title "dry metamorphism" is misleading. The section describes the "dry compaction" but not the metamorphism.

L214: tau is a threshold deviation, if I understand correctly.

L322: this is an incomplete sentence.

L397: avoid jargon "L-BFGS-B" and "bobyqa" and name the methods, references needed if you do not describe them

L464: "its" instead of "it's"

L464: how significant are the trailing decimals of 0.0299 if your search range was [0.01 0.2]? this suggests a precision that you presumably do not have.

L500: to start the model validation with questioning the observations is inappropriate! Describe the behavior of your model as exposed to observations. If applicable, quality issues with data may be discussed, but sub-ordinate to evaluating the model.

L522ff: avoid jargon "Pi16, Gu19; Pi16cal, Gu19cal, Jo09, . . .." in the text. The abbreviations are used to label plots and should be explained in the associated figure caption.

---

## Short Comment (SC2) · 5 Jun 2020

Dear Michael,

thanks for the answers.

I agree that my suggested cross-validation involves quite some (computational) effort, and its benefit is unknown; but at least you could get an idea for the influence of your SWE samples.

Nonetheless, for the "normal" validation, like you did with a holdout sample (even and odd years), it is considered best practice to use all your data for both training and

validation. In your case of a two-fold split, this would have meant repeating the fitting and validation also for the other variant (A: use even for training, odd for validation, B: use odd for training, even for validation). Maybe for the next time...

Best, Michael

---

## Author Comment (AC2) · 5 Jun 2020

Many thanks to Anonymous Referee #2 (AR2) for the praise and the recommendations to improve the paper. We will comment on them in the following.

(1) Thanks for describing this source of uncertainty, which we were not really aware of. However, for both datasets used in the study we cannot really allow for this source of uncertainty. For the Austrian measurements the respective snow depth values from the $SWE$ observations coincide with the associated value from the daily snow depths record. Swiss data (Marty2017, https://www.envidat.ch//metadata/gcos-swe-data) are provided as biweekly $SWE$ observations. Only those daily snow depth measurements

coinciding with the biweekly values were used. This is already described in the paper at L387ff. However, we will add a comment concerning this issue and name the omitted/accepted records.

(2) Thank you, you are right. We will add a note, that $SWE_{pk}$ is defined as the highest *measured* value, but not necessarily the maximum $SWE$ that occurred in a certain season.

(3) We will add information on how errors differ for shallower and deeper snow packs, perhaps for classes if there are enough readings to justify a statistic. We argue that absolute errors are more meaningful than relative numbers, which are extremely influenced by the denominator.

(4) Thank you for the references, we were not aware of the first one and will add parts of its results. As far as the second study is concerned we soon get into trouble not to compare the incomparable. The paper does not provide distinct numbers for *certain* models, but is a great (maybe the best) survey on how different models perform under different conditions/at different locations. We do not think that it is reasonable to compare our model to the mean bulk performance of other models, but we will add some numbers to the revised manuscript. As far as the third reference is concerned, we ignored it in favor of the more optimistic (from the SNOWPACK point of view) study of Langlois2009. We will add more numbers in the revised version.

Specific comments:
L20: Changed as suggested. See also our answer to AR1's comment (AC1).
L21: $SWE$ is an "areal density", not a (volumetric) density.
L69: See above. We will try to add results of the two suggested studies. We have not been aware of the second study. Thanks to AR2 for the notification.
L120: We will switch to snow depth throughout the paper. This will also change the title. Confer answer to AR1's comment (AC1).
L124: This will be considered in the text.
L159: We will switch to new snow throughout the paper.
L306: Changed to "model snowpack" in quotes.

L309: The value $450\,\mathrm{kg\,m^{-3}}$ was chosen arbitrarily as an example and is not connected to Rohrer and Braun (1991). By the way we are not aware of a publication "Rohrer and Brown, 1991", we guess this should be 1994.

L355: The phrase is literally taken from Sturm and Holmgren (1998) and is marked as such. However, it will be changed to "viscosity at [which] $\rho$ equals zero" (see also comment AC1).

L364: The sentence will be changed to "The last two parameters, $c_{ov}$ and $k_{ov}$, determine..."

L503: Following your comment and also one of AR3 we will subordinate this paragraph within this section and add a reference to point (1) above.

L506: This is of minor importance and will be omitted.

L576: An explanation of "diverse measurements" will be given in the revised manuscript.

L581: "now" will be omitted.

L582: All colors in Figs. 1 and 5 are the same, except for the "Scaling module". This is a black arrow in Fig. 1, whereas it is white with a black border in Fig. 5. We think that black bars in Fig. 5 would be distracting and therefore do not change.

L614: No, the jumpy behavior was detected, but does not play a role, because it is simply very small.

L664: Thanks for the hint to this unclear passage. Only mean snow depth of SNOW-GRID is used as covariate in the spatial model. The smooth model uses $SWE$ simulated with the $\Delta$SNOW.MODEL. This will be changed accordingly.

L667: We will add a comment for the average reader.

L691: We will mention this restriction of the $\Delta$SNOW.MODEL. The $\Delta$SNOW.MODEL cannot allow for mass loss due to sublimation. We will remark that at proper location in the revised manuscript.

L708: We assume Leppänen(2018b) is meant to be changed. The suggested study is brand-new and was not finally published when the manuscript was submitted. Thanks to AR2's comment this now can be updated.

L732: As suggested, the sentence will be changed, but the word "gapless" will be avoided. See respective comment of AR3 and our answer AC3.

L768: We agree, this statement might be too strident. It will be changed to "Nevertheless, the synopsis of the $\Delta$SNOW.MODEL and measured data gives hints on two important variables in modeling alpine snow: Typical mean density for new snow (24 h) might be better assumed below often used $100\,\mathrm{kg\,m^{-3}}$..."

The restrictions mentioned by AR2 will be added in the text. Regarding the first, see above at answer to L691. The second, the neglection of small snow fall events, will be outlined in the revised manuscript too. Still, including those minor events would even strengthen the arguments towards assuming new snow density below $100\,\mathrm{kg\,m^{-3}}$.
* * *
[Figure]

---

## Author Comment (AC3) · 5 Jun 2020

Many thanks to Anonymous Referee #3 (AR3) for thoroughly reading the manuscript and providing valuable criticism. We will carefully incorporate the suggestions, eliminate errors and inadequacies, and we are sure to end up with a significantly improved paper. AR3's major point of criticism is that the paper was "not focused in its objectives". In some respect this argument overlaps with AR1's, in parts also with AR2's criticism. The authors are very motivated to solve this issues during the revision and come up with a more precise concept and motivation.

Like it was stated in the "Answer to Anonymous Referee #1's comment from 28 April"

(AC1) we still want to see the △SNOW.MODEL as a new method to simulate SWE, which combines parts of older approaches. For the revised manuscript we will better specify the way how this is realized. But of course we also want to compare it with existing methods and proof its promising performance. We are convinced that both – "a)" and "b)" in AR3's comment – are necessary for the paper and we want to satisfy both.

During the presentation of the method we will be more clear on (physical) processes and will follow AR3's suggestion to describe them in advance. This might also help to satisfy AR1's arguments on keeping apart model code and snow processes.

Our "allergic" (cit. AR3) appearance regarding the usage of temperature is twofold: Firstly, using temperature would be nothing new at all. Many snow models could be run with temperature *and* snow height as input (and parametrizations of other variables), but hardly any with snow height *alone*. Secondly, there are indeed many precious daily snow height records without respective temperature observations. Some of them go back to the 19th century and even today there are manual snow height observations made with no respective temperature measurement. Downscaling or interpolation of temperature is often tricky, e.g. in alpine valleys with frequent temperature inversions etc. The △SNOW.MODEL evolves from a project to renew the Austrian snow load standard where we rely on old records for extreme value analysis. We will clarify this for the revised manuscript.

AR3 (like AR1) is highly critical of the null-hypothesis of the paper. We will omit it.

For the revised manuscript we will review what clearly is a result and what is discussion. We want to keep best parameter choices in the results section since they are important *outcomes* of the study. Description of parameter sensitivity is shifted to the discussion section as well as the interpretation of the optimized values. We are not quite sure what is meant by "uncertainty due to model structure" (cit. AR3). In case uncertainties are meant, that arise due to "structurally" omitted processes (wind drift, sublimation, etc.): Also AR2 criticized this aspect. There will be a respective statement in the revised

version. However, we cannot quantify these uncertainties.

Many thanks for the hint that the manuscript "leaves the impression the usage of viscosity is a unique, ingenious feature of this model." (cit. AR3). This is not at all what we want to purport. Of course Newtonian viscosity is broadly used for dry snow compaction. We will state that more clearly.

The authors will try to avoid side remarks and parentheses during the revision.

More details:
L8: We agree and will clarify.
L29: changed accordingly
L44ff: We agree and omit the acronyms.
L49: Sorry and thank you. This is indeed wrong the way it is written in the manuscript. We are aware of $SWE$ from passive microwave radiometry, but it is not available for the local and point scale, operational for many years. That is the point we want to make here. We will clarify.
L75ff: We agree; this might be confusing. The respective sentence is omitted in the revised manuscript and a note is added earlier where "thermodynamic snow models" are introduced.
L158: We agree to AR3's argumentation, but the term "deformation strain" is used by Jordan2010. We will name it "deformation" for the revised paper.
L168ff: We agree and will change to "Dry Compaction". Thank you for that. There will be a table in the revised version which clarifies the relation between modules and physical processes.
L214: This is true. $\tau$ will be introduced accordingly in the revised manuscript.
L322: "..." is changed to "etc."
L397: The acronyms of the methods are better known than their written-out names. Therefore, we keep them but provide the respective references in the revised manuscript.
"L-BFGS-B" is a limited memory (L) quasi-Newton method with the capability of handling bounds (B) of Byrd et al. (1995). BFGS stands for Broyden–Fletcher–Goldfarb–Shanno algorithm. "Bobyqa" implements optimization by quadratic approximation. The name is an acronym for Bound Optimization BY Quadratic Approximation (Powell, 2009).

L464: changed accordingly

L464: Rather certainly, but "presumably" (cit. AR3). Still, for consistency reasons we chose to provide three significant figures for all our optimized parameters. We change to two significant figures for the revised manuscript (except for $\rho_{max}$).

L500: We agree and will change that accordingly.

L522ff: Changed accordingly.

---

## Author Comment (AC4) · 5 Jun 2020

Dear Michael Matiu,

Many thanks for your comment on our publication. In the following we will answer to it step-by-step.

(1) R-package: This is addressed in the description of the function *nixmass* which comes with the package. The code is very slow if more than one year of continuous snow depth data is used as input. This is due to predefined matrices of the snowpack with all possible layers and days. In a future implementation this will be significantly

speeded up by iterating always only over two days, thereby avoiding pre-allocation of matrices.

(2) Structure: This criticism overlaps with comments of anonymous reviewers #1 and #3. It will be carefully considered in the revised paper.

(3) Validation method: In general we agree with your suggestion for a more sophisticated validation approach. On the one hand, this would provide uncertainty estimates for each parameter. On the other hand, one could not arbitrarily choose parameters within the validated ranges, because optimized parameters interrelate. The parameter estimates would not benefit from a cross validation approach, where you end up with a likely range for each parameter. Which parameter values should one choose for running the model? We do not see a real benefit for the application of the model, since you have to use a set of optimized parameters.
An uncertainty assessment of more practical use is already presented in the study. We show the sensitivity of simulated $SWE$ values to changes of input parameters. Not least, a cross-validation procedure as suggested by you, requires about one week of optimization time. We argue, that it is not worth the effort and will not implement this recommendation.

---

## Author Response (AR1)

Dear Editor and Referees!

The authors want to thank you for your valuable comments and suggestions. We are sure they led to an improvement of the manuscript.

Referee comments RC1 to RC3 as well as short comment SC1 were answered in detail during the discussion process already (see AC1-4). Now we provide the revised paper as a fluently readable document and as a version where all changes are marked and commented (see below). All specific referee comments were considered according to AC1-4. In addition, all general criticism is addressed and the paper was improved accordingly:

- The paper was restructured and sharpened. Its aim – introducing and evaluating a new method to model *SWE* – was emphasized.
- The placement of the model within the snow modeling spectrum was specified, not least by providing Table 1.
- The entangled descriptions of model code, modules and snow physics were clarified, not least by including Table 2.
- Language and writing were improved; repetitions eliminated and parentheses avoided.
- Requested citations were analyzed and included, whenever possible.
- More details on the data used for calibration and validation were given. A respective table was added in the appendix (Table A1).
- The application example was moved to the appendix.
- Discussion-like parts from the initially submitted manuscript's Results section were moved to the Discussion section.
- Main text was shortened by ca. 3000 words (>20%).

We updated the results of our error analysis, initiated by RC2 but without providing relative numbers (see AC2). We now provide RMSEs for all *SWE* and $SWE_{pk}$, as well as gradings depending on absolute *SWE* (cf. Fig. 2, Table 4 etc.), instead of statistics of RMSEs from different stations. This changed some key values concerning model accuracy, but none of the overall statements. Revisiting data sources and providing Table A1 revealed some minor inconsistencies concerning the number of observations and years, which could be solved easily.

We hope the revised manuscript fulfills HESS journal requirements for a final revised paper, and we are looking forward to your answer.

With best regards,
Michael Winkler, Harald Schellander, Stephanie Gruber

**Snow Water Equivalents exclusively from Snow Depths and their temporal Changes: The $\Delta$SNOW.MODEL**

Michael Winkler[1,*], Harald Schellander[1,2,*], and Stefanie Gruber[1]

[1]ZAMG – Zentralanstalt für Meteorologie und Geodynamik, Innsbruck, Austria
[2]Department of Atmospheric and Cryospheric Sciences, University of Innsbruck, Austria
[*]These authors contributed equally to this work.

**Correspondence:** Michael Winkler (michael.winkler@zamg.ac.at) and Harald Schellander (harald.schellander@zamg.ac.at)

**Abstract.** [1]Snow depths[2] have been manually observed for many years, sometimes decades, at various places around the globe. These records are often of good quality. In addition, more and more data from automatic stations and remote sensing are available. On the other hand, records of snow water equivalent $(SWE)$ – synonymous for snow load or mass – are sparse, although it might be the most important snowpack feature in  hydrology, climatology, agriculture, natural

5   hazards research, etc. $SWE$ very often has to be modeled, and respective models either depend on meteorological forcing or are not intended to simulate individual $SWE$ values, like the substantial seasonal "peak $SWE$".

    The $\Delta$SNOW.MODEL is presented as a new method to simulate local-scale $SWE$. It solely needs a regular time series of snow depths as input. The $\Delta$SNOW.MODEL is a semi-empirical multi-layer model and freely available as R-package. Snow compaction is

10   modeled following the rules of Newtonian viscosity. The model considers measurement errors, treats overburden loads due to new[3] snow as additional unsteady compaction, and melted mass is stepwise distributed top-down in the snowpack.

    Seven model parameters are subject to calibration. Snow observations of 67 winters from 14 stations, well-distributed over different altitudes and climatic regions of the Alps, are used to find an optimal parameter setting. Data from another 71 independent winters from 15 stations is used for validation.

15    Results are very promising: Median bias and root mean square[4] error for $SWE$ are only $-3.0\,\mathrm{kg\,m^{-2}}$ and $30.8\,\mathrm{kg\,m^{-2}}$, and $+0.3\,\mathrm{kg\,m^{-2}}$ and $36.3\,\mathrm{kg\,m^{-2}}$ for peak $SWE$, respectively. This is a major advance compared to snow models relying on empirical regressions and even sophisticated thermodynamic snow models do not necessarily perform better.

20
* * *
[1]Changes based on specific and general reviewer comments are marked in blue and orange, respectively.
[2]Snow "height" was changed to snow "depth" throughout the paper and also in the title. The respective symbol $H$ was changed to $HS$, and $h$ was changed to $hs$.
[3]"Fresh" snow was changed to "new" snow throughout the paper.
[4]Root mean "squared" error was changed to root mean "square" error throughout the paper.

**1 Introduction**

[revised manuscript text omitted]

* * *
[15]Figure A1 and Table A1 were added during the revision. They were not part of the primarily submitted manuscript.

[16]The following remarks on $SWE$ observation accuracy were placed at the beginning of the Validation and Illustration section of the initially submitted manuscript and moved to this place during revision.

Model calibration was performed with the statistical software *R* (R Core Team, 2019) and the *R* package *optimx* (Nash, 2014). Results were obtained with optimization methods *L-BFGS-B* (Byrd et al., 1995) followed by *bobyqa* (Powell, 2009), which both are able to handle lower and upper bounds constraints. The function to be minimized was the root mean square error (RMSE) of $SWE$s from the $\Delta$SNOW.MODEL and observed $SWE$s, using the calibration data set $SWE_{\mathrm{cal}}$.

**3 Results**

The following evaluates the ability of the $\Delta$SNOW.MODEL to calculate snow water equivalents exclusively from snow depths, and its practicability.[17]

**3.1**

Table 3 gives an overview of all parameters and summarizes the optimal setting for the $\Delta$SNOW.MODEL. A discussion of the best-fitted values and of the model sensitivity to parameter changes can be found in Sect. 4.

The minimal RMSE between all $SWE$ observations used for calibration ($SWE_{\mathrm{cal}}$) and the respective modeled values is $30.1\,\mathrm{kg\,m^{-2}}$. It is reached for new snow density $\rho_0 = 81\,\mathrm{kg\,m^{-3}}$, maximum density $\rho_{\mathrm{max}} = 401\,\mathrm{kg\,m^{-3}}$, "viscosity parameters" $\eta_0 = 8.5 \times 10^6\,\mathrm{Pa\,s}$ and $k = 0.030\,\mathrm{m^3\,kg^{-1}}$, threshold deviation $\tau = 2.4\,\mathrm{cm}$, and "overburden parameters" $c_{\mathrm{ov}} = 5.1 \times 10^{-4}\,\mathrm{Pa^{-1}}$ and $k_{\mathrm{ov}} = 0.38$.[18]

**3.1.1**

~~Being aware of both – the huge possible variations of new snow density $\rho_0$ depending on meteorological conditions during snowfalls and the possible cruciality of this parameter for $SWE$ simulation by the $\Delta$SNOW.MODEL – $\rho_0$ was chosen to be a constant in the framework of the model. For $\rho_0 = 81\,\mathrm{kg\,m^{-3}}$ the minimal root mean square differences/errors (RMSE) between all $SWE$ observations used for calibration ($SWE_{\mathrm{cal}}$) and the respective modeled values was reached. This value clearly lies within the broader frame of possible new snow densities and quite closely to Sturm and Holmgren (1998)'s $75\,\mathrm{kg\,m^{-3}}$ (The $\Delta$SNOW.MODEL could be seen as an extended combination of the Sturm and Holmgren (1998) and Martinec and Rango (1991) approaches.), but it is found in the lower part for "typical" new snow densities (e.g., Helfricht, 2018). A possible explanation could be that the $SWE$ measurement records used for the calibration tend to underrepresent late winter and spring conditions. Regular (weekly, biweekly) observations capture the short melt seasons worse than the (much) longer accumulation phases.~~
* * *
[17]Most content of subsections 3.1.1 to 3.1.5 of the submitted manuscript was transferred to the Discussion section during revision. The segmentation 3.1.1-3.1.5 was omitted.

[18]All the optimized values (except $\rho_{\mathrm{max}}$) where rounded to two significant figures for the revised manuscript.

[revised manuscript text omitted]

---

## Referee Report (RR1)

[referee-annotated manuscript omitted]

---

## Author Response (AR2)

Dear editor Prof. Dr. Markus Weiler,

We want to thank you for thoroughly reading our revised manuscript (RM1). Indeed, the two reviews are "quite different". We, the authors, would say they are contradictory: Reviewer #1 admits some improvements but basically repeats arguments from the discussion phase, although they were addressed (i.e., changed or answered) during the revision. Reviewer #2 grades the paper as "excellent" in significance and quality. Being aware of this complicated situation, we regret your decision to solely follow reviewer #1's argumentation.

You highlighted four points, which we address in the following. Afterwards we comment reviewer #1's statements. The revised manuscript (RM2) was uploaded to Copernicus Publications. We followed most of the suggestions and again made a lot of changes between RM1 and RM2 (shown in green below and marked in the track-change file), but we want to emphasize that none of these changes touches the core of the paper, since Reviewer #1 does not have concerns about the model architecture, the method, the calibration, the validation, the results, the discussion, the plots, the tables.

We are sure the manuscript was improved during this second major revision and want to thank you and the referees for your suggestions. Changes in style, wording and language are partly significant and consulting a native speaking professional was very helpful. As there was no more criticism on the content, the method and the model as such, we are confident that this work now fulfills the high standards of HESS.

Sincerely,

Michael Winkler, Harald Schellander, Stefanie Gruber
* * *
Answers to editor points 1) to 4):

1) "The language is still difficult - please try to find a native English speaker to improve the language".
   The manuscript was given to a native-speaking professional. All her suggestions were implemented in RM2. We thank for this work in the acknowledgements.
   Important changes suggested by the native are: (1) the re-writing of the abstract and (2) moving the paragraph on the importance of SWE from the end of section 1.1. to section 1 (right after equation 1).

2) "Length: the paper and in particular the method section is too long. The method section extends over 8 pages - very detailed, but too detailed for a scientific paper. This should not be a report to an agency. Please constrain the method section much more and refrain from writing the section like a code. Describe what the models does, what physical equation you use, but not how these are taken in the code. If you think this is really necessary, a flow chart with the main model parts would be much more helpful."
   We addressed your critics on "writing the section like a code" and could shorten the methods section for RM2 by 20%. We believe that an even greater shortening would lead to a poorer understanding of the text. Two comprehensive equations could be omitted. We also shortened passages from other sections, skipped repetitions and avoided accentuations.
   RM2 is about 1000 words (about 10%) shorter than RM1. It is about 21 pages in HESS's final two-column version incl. 6-7 pages of abstract, references, appendix etc.

3) "No selling of the model: there is no need to sell the model to the scientific community. Either it is good, provides new ideas and is superior to other models or it is not. So please refrain from sentences like this: "seems to be the first model since long that ..... Not particularly innovative, but remarkably successful...." This is not a good scientific style."
We avoided expressions awaking the impression of selling our work.

4) "I also think that including temperature or other meteorological variables into the model framework may further improve the model. Measurement of precipitation is certainly difficult and hence a model driven by snow depth makes a lot of sense. But how much could the model improve if these observations would be considered. This would be scientifically very interesting. If you can show, that this would not improve the model much (at least showing it at one location), this would certainly strengthen the realist of your study."
Including temperature etc. would go directly against what we want to present with the ΔSNOW model for the reasons given in the abstract and introduction. The model is intended to run without any additional meteorological input to make efficient use of long term snow depth series not accompanied by e.g. temperature recordings.
The new model was intensively compared to other models that need no further meteorological input. Its accuracy was also compared to the respective accuracy measures that are available for more complex models. Running thermodynamic models of several complexities aside the ΔSNOW model and compare their results for SWE would be an interesting experiment for the future. However, it would burst this paper's limits and would definitely deserve a separate journal article.
* * *
Reviewer #1's letter and our answers and comments.

Dear authors,
the manuscript has profited from the improvements you have realized.
We agree and thank for the elaborate comments.

However, several important issues still deserve attention, some of which have been addressed in the first round already, but they are not sufficiently solved. These issues should be seriously elaborated on prior to publication of the manuscript as full paper.
This is not true. *All* issues have been addressed and seriously elaborated (see author comments of the discussion). The vast majority of the reviewers' comments led to changes, some issues were defended. "Sufficiently solved" is a very subjective expression (cf. reviewer #2).

(1): it is still not entirely clear what the ultimate aim of your presented work is.
We changed and sharpened the wording of the Introduction, especially of the motivation section 1.3, which outlines the paper's aim. We also moved a paragraph from section 1.1 to the opening section to provide more of a "set up" for the whole paper.

Main points: (i) You say you „present a new model, a new method to simulate SWE."
The above mentioned phrase is not used in RM1. We avoided the terms "model" and "snow model" in RM1. Following reviewer comments of the discussion phase we declared, that we present a "new method". We reconsidered the issue and stepped back for RM1, see below.

Usually a snow modeller would expect a "simulation of SWE" to be based on meteorological forcings (whichever, depending on the empirical or more physical type of the model).
We cannot agree on that. There are snow models based on meteorological forcings, we clearly define

them as "thermodynamic models" (like Langlois et al., 2009), and there are snow models that rely on snow depth and non-meteorological variables like date, altitude etc. The latter we define as "empirical regression models" (ERMs; in the tradition of Avanci et al., 2015) since those are *only* depending on regressions, but do not resolve any thermodynamic processes. The third category that we identify, are the "semi-empirical snow models" (following Gruber, 2014), where the ΔSNOW model belongs to. The latter do not resolve thermodynamic processes, but make use of basic rules of densification. They are not "fully" but "semi"-empirical, since these densification rules are part of theoretical physical concepts like, e.g., Newtonian viscosity. These definitions are very well described in RM1, esp. in Table 1, but we have again sharpened the wording for RM2 and introduced subsections for each model type for better differentiation.

If a "snow modeler" would only expect a meteorological-forcings-based model for a "simulation of SWE", he/she ignores the whole bunch of ERMs and semi-empirical models, but is only open for thermodynamic models. What else than "simulation of SWE" do all ERMs and semi-empirical models (like references in *) do?

What you present to the reader effectively is a layered snow density evolution parameterization for ephemeral snow packs, from which - of course - swe can be derived. Maybe this could be expressed more precisely.

We identify a misunderstanding: Reviewer #1 seems to only count thermodynamic models as legitimate snow models. ERMs and semi-empirical models are claimed to be "density parametrizations". This is different to what was done in the past. Many ERMs and semi-empirical models have been presented as "models". Yes, one could argue that ERMs are only parametrizations of density (or SWE) and do not deserve being called snow models. Still, this is in contrast with authors like Jonas et al. (2009) or McCreight and Small (2014), who naturally call their approaches "models". But semi-empirical models are definitely more than pure parametrizations since they use, e.g., viscosity physics etc. Why should the ΔSNOW model only be a "layered snow density evolution parameterization"? We already called it (only) a "new method" in RM1 rather than a "new model" as we followed former reviewer comments. In this respect this was more modest than most presentations of ERMs, although our method is more complex.

Having revisited this issue, we now stepped back and again call the ΔSNOW model a "model" rather than a "method" for RM2, since also less complicated approaches have naturally been called models in the past. Maybe it seems offensive to have the word "model" in our initial model's name (ΔSNOW.MODEL). We now simply call it "ΔSNOW". (This slightly changes the title of the paper.)

(ii) There is still no strong argument to resign from using available meteorological observations. Even if your goal is to develop an assimilation scheme that uses nothing more than recorded HS, a comparison of this scheme to a model using at least temperature records (generally available, extendable with humidity for phase detection) would be scientifically comprehensible approach.

Isn't the lack of meteorological observations (aside snow depth) a *very* strong argument? Are all the other papers (*) on ERMs and semi-empirical models no "scientifically comprehensible approaches"?

There should be locations where even precipitation is available, allowing for extending the classical temperature index approach (and, e.g., including rain-on-snow). However, I do agree that this is an option - but a crucial one, as long as you call your model a snow model.

1) "as long as you call your model a snow model": We actually did not call it a snow model in the RM1, but we now call it a model in RM2 again, since also much simpler approaches have been called models in the past (see above).

2) There are only very few places in the Alps with regular SWE measurements *and* daily HS measurements *and* temperature/precipitation recordings for many years. It was quite hard already to put together those 6 Austrian and 9 Swiss stations with SWE and HS

measurements. We emphasized this even more for RM2 (section 2.3.1). The fact that the Δsnow model requires no other input than snow depth makes it also very worthy to derive SWE from remotely sensed HS recordings. This aspect was also emphasized for RM2.

3) Of course temperature, precipitation etc. could be included in the ΔSNOW model. However, this step would be nothing innovative. Many thermodynamic models can be run with HS, temperature, precipitation (and parametrizations of anything else). First of all: Why should one try to improve a method that should work without a certain input by using this respective input? That makes no sense.

(iii) As the paper is now, it seems that the ultimate goal of your numerical experiment was the snow load map, to be developed without meteorological recordings (because they are not available everywhere for the entire period). If this is the case, the paper should better put this fact in the focus and follow a respective structure and argumentation. If your goal is the model development and comparison with its peers, as you present it in the validation/comparison section, you should focus on the latter and remove much of the rest that does not contribute to this goal. Please make the paper as much as possible clear in its intention.
This criticism was addressed in the revision already. RM1 is about a method to calculate SWE. In the appendix an example of application is given by providing the snow load map.

(2) I still strongly recommend to separate any mathematical formulation of empirical/physical relations from the implementation of the model (code, modules etc.). Swapping two different types of elements in the model along the model description is more puzzling than helpful. Better present the equations of the model in the text of the manuscript, and the code (with its structure – „modules" in your particular implementation) as flow process chart in the appendix.
We made comprehensive changes for RM2 addressing this, but still permit ourselves to provide the mathematical formulations of empirical/physical relations implemented in the model.

(3) The language still is common speech like in many formulations, e.g. „a bit of dexterity" (39), "with the help of De Michele (2013)" (212) or „For the ΔSNOW.MODEL this kind of high error tolerance of ρ0 is a rather feeble argument to use a power law" (202/203). Also better avoid all formulations where the model, the code or a process „does" or „decides" or „ignores" something; It is the modeller who does or decides or ignores. I generally recommend professional native English language support.
We changed accordingly for RM2, and engaged the service of a native speaking professional.

(4) The manuscript should be significantly shortened.
The manuscript was already shortened by 3000 words after the discussion. RM1 is about 24.5 pages in HESS final two-column layout. We could further shorten the text for RM2, mainly in the method section. RM2 should be about 21 pages.

There are entire paragraphs with long explanations that could be replaced by just presenting the mathematical formulas with the respective references (e.g., 384-404).
This paragraph (lines 384-404) is key. How can one calibrate physical parameters without defining their potential limits in advance? Furthermore, these paragraphs are the base of the classification provided in the results section.

And, if you have shaped the focus of the experiments, much of the rest can be omitted. Examples: (i) there is no need to argue the advantages of HS measurements against anything else (chapter 1.1) - long-term observations are always valuable.
Chapter 1.1 is about the big discrepancy between HS and SWE measurements, which is worth being addressed in this context.

We strongly shortened this paragraph for RM2.

(iii) 349-351: it seems that what you call „scaling" is sort of an assimilation of the HS observations into the current model state by means of an replacement/adjustment of respective variables. Why not say this this with few sentences and skip the rest?

We could only slightly shorten this section. The way how the "scaling" is done is crucial, novel, innovative, not at all straight forward (see App. B), and therefore worth describing.

Other details I came across (exemplary, several occurences of the respective type in the text):
- 89ff: your division of SWE models in the „thermodynamic" and „empirical regression" type of model is unusual (more usual are energy balance vs. temperature-index). Maybe it should clearly be indicated here that you use the expressions for density models?

What we mean with "thermodynamic", "empirical regression" and "semi-empirical" models is clearly defined in the manuscript and supported by references, e.g. Langlois et al. (2009) or Gruber (2014), see above. In many other contexts the differentiation between energy balance and temperature-index models is necessary (In this respect it is "more usual".), but energy balance and temperature-index models are both encompassed by "thermodynamic" models. However, we need the cutting line between "thermodynamic" and "empirical regression" models here.

- entire chapter 1.2: this chapter should lead to the next one, the motivation. However, a clear argumentation (following the Rango/Martinec model description) of what is missing there, or what could/should be improved, is not given. The last sentence seems to loosly tight to what was stated before

We sub-divided the chapter in three parts for RM2. The argumentation is given in the following motivation section (1.3), which was sharpened in this respect. We also shifted the last sentence of 1.2 to 1.3.

- 194: „For non-zero snow depth observations": how many?

We do not understand the question. This is a description of the model concept. It is irrelevant how many non-zero snow depth observations there are. The model works for ephemeral snow packs lasting, e.g., three days as well as for, e.g., high-altitude snow packs lasting many months. We estimate a limit of 200 days in the manuscript.

- 220: „constituted by the sum of loads overlying layer i": something is missing here to make the sentence complete

The sentence is correct. We changed "constituted" to "induced" for RM2.

- 217, 224: avoid „today's …" (better „the most current" or the „actual")

We changed to the suggested version.

- 230: „cannot be observed": why? Did you make an attempt to illustrate density recordings and analyze if and when these reach a maximum value?

No, we did not illustrate density recordings here, but many other did before. There is no "natural constant" that defines a seasonal bulk density maximum, but it is a model constant. That's what we wanted to express here. We removed this phrase.

- 237: better formulate that the age of the layer is a surrogate for the physical processes and their evolution and effect over time

Changed to "Equation 2 links the densification rate to the layer age, but indirectly by the use of density, and not directly as it was the case with Martinec and Rango (1991)'s power law approach.

Consequently, ΔSNOW's compaction is not directly depending on layer age, which is a prerequisite for the functioning of the *Overburden Submodule* (Sect. 2.2.1)"

- 287-290: this certainly depends on the time step of the model
Yes. As this statement is already in brackets we avoid being even more precise here.

- 325: „in the model world": what is a „world" here?
We avoid this for RM2.

- 335: „a small "stretching" of the snowpack is necessary": better avoid such formulations: there is no stretching, but a modification of state variables
The wording is changed for RM2, as other pictorial expressions (see above). Still, "stretching" is presented/defined here as the opposite of compaction. Compaction is (naturally) accepted as an expression, therefore also "stretching" must be acceptable, even more as it is put in quotation marks.

- 387: „Sub-daily means of new snow densities are lower": when and why? Please explain
See respective reference.

- 398: „7.62237 × 106 Pa s": the number of decimal places should be reduced to one
This is a literal quote. It is clearly marked as such and cannot be changed. For the values we found, we use "two significant figures" (except for $\rho_{max}$). See AC3 of the discussion phase.

- table A2: correct „valibration"
Changed accordingly for RM2.
- etc. …

* Guyennon et al. (2019), Pistocchi (2016), Gruber (2014), Sturm et al. (2010), Jonas et al. (2009), Mizukami and Perica (2008), McCreight and Small (2014), Martinec (1977), Martinec and Rango (1991), Rohrer and Braun (1994), and many others (see e.g. Avanzi et al., 2015)
* * *
Changes from RM1 to RM2 that are not related to referee and editor suggestions:

- Capital letters are omitted in title and section names (following a native's suggestion).
- The abstract was rewritten.
- Citation Paterson (1998) was changed to newer Cuffey and Paterson (2010).
- In section 4.8 an outlook was given citing Lievens et al. (2019).

---

## Author Response (AR3)

Dear editor Prof. Dr. Markus Weiler,

We again thank for guiding the process and appreciate your acknowledgement regarding the studies relevance.

As suggested, we removed Eq. 1 and edited the beginning of the introduction.

Sincerely,

Michael Winkler, Harald Schellander, Stefanie Gruber